# WaterMax: breaking the LLM watermark detectability-robustness-quality trade-off

**Eva Giboulot**
Inria, CNRS, IRISA
University of Rennes
Rennes, France
eva.giboulot@inria.fr

**Teddy Furon**
Inria, CNRS, IRISA
University of Rennes
Rennes, France
teddy.furon@inria.fr

## Abstract

Watermarking is a technical means to dissuade malfeasant usage of Large Language Models. This paper proposes a novel watermarking scheme, so-called WaterMax, that enjoys high detectability while sustaining the quality of the generated text of the original LLM. Its new design leaves the LLM untouched (no modification of the weights, logits, temperature, or sampling technique). WaterMax balances robustness and complexity contrary to the watermarking techniques of the literature inherently provoking a trade-off between quality and robustness. Its performance is both theoretically proven and experimentally validated. It outperforms all the SotA techniques under the most complete benchmark suite.

## 1 Introduction

The availability of powerful large-language models (LLMs) allows users to produce texts that look like human writings. The risk for misuse of these models is critical, ranging from the impersonation of individuals to the large-scale generation of fake news. Identifying the provenance of a given piece of text is paramount to limit the impact of such 'weaponization' of LLMs. New initiatives or regulations impose technical means for AI traceability [28, 9, 5].

Forensics passive methods generally leverage a priori knowledge about the statistics of texts generated by a given class of LLMs [34, 27]. Despite their versatility, these methods offer low performance. The reported probabilities of errors are only validated empirically on some datasets, and because of this, they are never lower than $10^{-3}$ [14].

In contrast, active methods like watermarking are only limited by the fact the LLM owner must integrate the watermarking within the generation processes. This is done by embedding an imperceptible signal in the generated text, which can be retrieved by a detector sharing the secret key of the model owner. Current watermarking methods for generative texts [22, 1, 23, 8] provide low and guaranteed false positives rates. Yet, the trade-off between the detectability and the text quality crucially depends on the entropy of the text to be generated, which in turn depends on the prompt and the LLM, as illustrated in Fig. 1. This implies that the distortion-free property [1, 23, 8] ensures that watermarking does not degrade text quality but inherently limits the detectability.

This paper presents WaterMax, a watermarking technique that trades off robustness not for quality, but for complexity. It obeys the regular constraints found in the literature: it can be integrated into any standard LLM without fine-tuning the weights, the detection does not need the original LLM, and it can spot the watermark on a slice of generated text with some guarantee on the false positive rate. Our contributions are the following:

38th Conference on Neural Information Processing Systems (NeurIPS 2024).

- WaterMax is based on a new design not relying on the usual mechanisms of the literature; especially, it keeps the next token distribution and sampling (temperature and method) intact. Moreover, it better utilizes the text entropy by working over sub-sequences of tokens, so-called chunks hereafter, rather than token by token.

- This new design makes WaterMax almost distortion-free, as shown experimentally (Sect. 7.3) and justified theoretically (App. H), and yet enjoys higher robustness than the state-of-the-art, consistently over several LLMs. Figure 1 shows that the other methods need to boost the watermark strength to be as detectable as WaterMax, inevitably degrading the quality.

- This new design facilitates building up a theoretical model of the watermark robustness characterizing the true positive rates under attack (see Prop. 5.2).

## 2   Related Work

Watermarking texts generated by LLMs is mainly performed by changing either the distribution [21] or the sampling [1, 23, 8] of the next token. Critically, the false-positive rate of these methods can be reliably controlled [11]. Furthermore, the computational cost of most of these methods is negligible relative to the text generation itself. On the other hand, they all suffer from similar weaknesses:

**Text entropy limit**   Some theoretical works [8, 17, 21, 1] show that the watermark detectability is highly dependent on the entropy of the text to be generated. In practice, this makes the text length necessary for reliable watermark detectability dependent on the type of the text, and also the LLM. One may increase the watermark strength or the temperature in the LLM to increase the entropy artificially [19, 26]. Both solutions degrade text quality compared to the original LLM.

**Distortion-freeness and quality**   Kuditipudi et al. [23] define that a watermark is distortion-free if it does not modify the probability distribution of the next token *on average* over the keys. This is guaranteed in schemes like [1, 23, 8]. Yet, these schemes rely heavily on the entropy of the token distribution, leading to possibly low detectability without any means to increase it without losing distortion-freeness. Appendix J illustrates the dependence of Aaronson's scheme on the original LLM. Moreover, a scheme that is not distortion-free does not necessarily lead to texts with lower *empirical* quality, but foreseeing the impact of the watermark strength on the quality is an issue.

**Watermark robustness characterization**   The watermark must resist text editing ranging from a simple insertion of words to a complete paraphrasing of the text. At the time of writing, there are only two watermarking schemes designed from the ground up to be robust [23, 35]. Yet, other state-of-the-art methods have experimentally shown resiliency to attacks against long-form texts [29] under a precise control of the false-positive rate [11]. However, none of these methods can theoretically guarantee its robustness. Only bounds on the moments of the detection score are provided in [21] and [1].

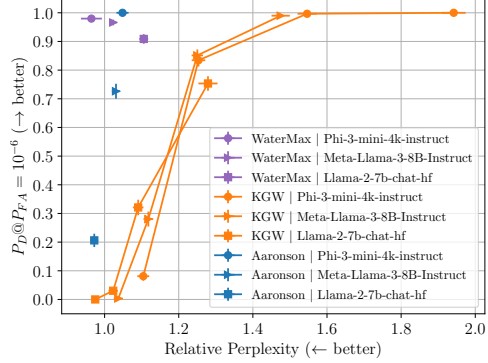

Figure 1: Detectability as a function of text quality for different LLM architectures. WaterMax always reaches a detectability close to 1 despite a negligible loss of quality. Probability of false-alarm fixed at $10^{-6}$, nucleus sampling ($top_p = 0.95$) at temperature 1.0. Text quality is measured as the relative perplexity of the watermarked text over the non-watermarked text.

This work presents a scheme that **empirically** incurs a negligible loss while reaching an arbitrarily high detectability even on small texts. The parameter tuning trades robustness for complexity but has almost no impact on the text quality. Moreover, we characterize its performance under a large attack range by expressing both false-alarm and true-positive rates. Its drawback is a large computational cost, which we show how to limit throughout the paper.

## 3 Main Idea: Watermarking by generating multiple texts

This section first presents a simplified and inefficient version of the method for a pedagogical purpose. The driving idea is that a LLM is a randomized algorithm. In its simplest implementation, the LLM computes a probability distribution of the next token and *randomly* samples according to this distribution. The chosen token is appended to the context, and the process iterates. Our main idea is to let the LLM generate several texts for a given prompt and select the one that is the most suitable from the watermarking point of view.

Initially, we select a sound watermark detection algorithm in the sense that it can output a $p$-value for any piece of text under scrutiny. Usually, such detection computes a score and then 'normalizes' it in a $p$-value defined as the probability that a non-watermarked text yields a score equal to or higher. Appendix A lists some candidates from the literature and presents our own design.

Our watermarking embedding lets the LLM generate $n$ texts for a given prompt. Since all pieces of text are generated normally, *i.e.* without being degraded by a watermark, their qualities are likely high. From the $n$ generated texts, the LLM outputs the text with the lowest $p$-value.

The advantages are that the text quality is not degraded and that any sound detection may be used. The price to pay is high complexity and long latency. Suppose that a text is deemed watermarked if its $p$-value evaluated by the detector is lower than the required false alarm probability $P_{FA}$. This raises the question of the number $n$ of generated texts for successfully embedding the watermark.

**Proposition 3.1.** *The detectability measured by the power of the test, i.e. the probability $P_D$ of detecting a watermarked text is the following increasing function w.r.t. $n$:*

$$P_D \triangleq \mathbb{P}(P < P_{FA}|\mathcal{H}_1) = 1 - (1 - P_{FA})^n. \tag{1}$$

Appendix B gives a sketch of the proof. We point the reader to two important advantages:

- The performance does not depend on the choice of score function used to obtain the $p$-value.
- The performance does not depend on the length of the text. Consequently, even an extremely small text can be watermarked in theory.

Figure 2 plots the power of the test as a function of $n$ for various probabilities of false alarm. Whatever the choice of $P_{FA}$, one can clearly observe that a huge number of generated texts ($\gg 50$) is required to obtain a power greater than $0.5$, which is unacceptable in practice. Section 4 shows how to improve this base algorithm to reach arbitrarily high power with a smaller computational power.

Another weakness is the assumption that the LLM can create $n$ texts whose $p$-values are independent. For some prompts, the diversity (*i.e.* entropy) is indeed small, which implies that the LLM creates similar texts or even duplicates of text among the $n$ outputs. Section 6 investigates how self-synchronization mitigates this issue.

## 4 Watermarking chunks of text

This section devises ways to efficiently explore the space of possible texts to find a low $p$-value. The idea is to split the generation into $N$ iterations, each iteration creating a chunk of the text. The aim is to reduce the computational burden by generating small chunks while exploring many possible texts.

### 4.1 Exploring the text space

For each chunk, a batch of $n$ text drafts is generated independently based on the text candidates of the previous iteration. Ideally, one would generate $n$ drafts at each chunk for each previous candidate, resulting in a tree of $n^N$ texts from which to choose the lowest $p$-value. Obviously, this is not

tractable. We reduce the complexity by keeping only the $m$ best candidates at each step, akin to a Viterbi algorithm [33]. The pseudo-code is summarized in Alg. 1 in App. F. This is suboptimal because the candidates minimizing the $p$-value at a given step may not be the best once completed at iteration $N$. Other exploration strategies range from greedy search to Monte Carlo Tree Search [6]. The pseudo-code is summarized in Alg. 1 within Appendix F.

## 4.2 Cumulative scores

This paper considers detection schemes that share the following process. The vocabulary $\mathcal{V}$ is a list of $|\mathcal{V}|$ admissible tokens, and a text is decomposed into a sequence of $L$ tokens. Depending on the secret key $k$, the detection associates to the $i$-th token a variable $u_i$. The score is computed as the sum over the tokens $s = \sum_{i=1}^{L} u_i$. This is translated into a $p$-value assuming that, under $\mathcal{H}_0$, i) the scores of the tokens are independent and identically distributed random variables $(U_i)_{i=1}^{L}$, giving birth to a random score $S$, ii) whose c.d.f. $F_S(\cdot; L)$ is known so that the $p$-value is simply:

$$p = \mathbb{P}(S > s) = 1 - F_S(s; L). \tag{2}$$

Appendix A lists some detection schemes of the literature following this procedure. Our choice is simply $U_i \overset{\text{iid}}{\sim} \mathcal{N}(0; 1)$ so that $F_S(s; L) = \Phi(s/\sqrt{L})$.

For the sake of clarity, suppose that the chunks are composed of $\ell$ tokens. At iteration $i$, the $j$-th candidate has a score denoted $s_{i,j}$ and $n$ drafts for its following chunk are proposed. This produces $n$ incremented scores per candidate, thus $mn$ variables: $s_{i,j} + \delta s_{i,j,k}, 1 \le j \le m, 1 \le k \le n$. Since these scores are converted into $p$-values by the same function, *i.e.* $p = 1 - F_S(s; \ell(i + 1))$, selecting the $m$ lowest $p$-values amounts to keep the $m$ maximal cumulative scores, hence the name WaterMax.

## 4.3 Low latency

One issue is the latency: the final text cannot be issued until all candidates reach an end of sequence. This problem is fixed with the drastic choice $m = 1$. It amounts to a greedy search appending to the text at iteration $i$ the chunk draft yielding the biggest incremental score $\delta s_{i,1,k}$. This choice enables the output of a chunk of text at the end of each iteration, hence reducing latency. Of note, this also corresponds to the draft with the lowest local $p$-value $1 - F_S(\delta s_{i,1,k}; \ell)$.

## 4.4 Optimal detection without attack

This section introduces a simple detector to reveal the advantage of our watermark embedding. The idea is that the received text is composed of chunks whose local $p$-values are distributed as beta distributions $B(1, 1)$ (*i.e.* $\mathcal{U}_{[0,1]}$) under $\mathcal{H}_0$ or $B(1, n)$ under $\mathcal{H}_1$ (see Appendix. B). This is a generalization of the algorithm of the previous section, the only difference being that the likelihood

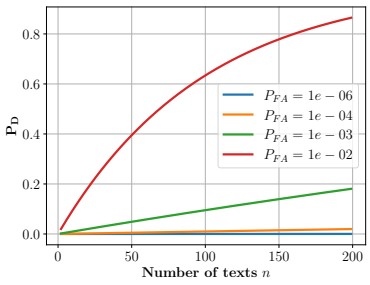

Figure 2: Theoretical power of the uniformly most powerful test (1) as a function of the number of generated texts $n$.

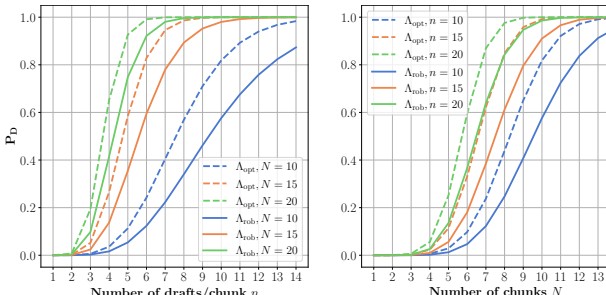

Figure 3: Theoretical power of the optimal (3) and robust 5 tests at $P_{FA} = 10^{-6}$ for $m = 1$ as a function of the number of drafts $n$ per chunk and the number of chunks $N$.

ratio (12) now aggregates a vector $\mathbf{p}$ of $N$ observed local $p$-values:

$$\Lambda_{\mathsf{opt}}(\mathbf{p}) = -\sum_{i=1}^{N} \log(1 - p_i) \lessgtr_{\mathcal{H}_1}^{\mathcal{H}_0} \tau. \qquad (3)$$

**Proposition 4.1.** *The optimal detector $\Lambda_{\mathsf{opt}}$ has the following performances when there is no attack:*

$$P_{FA} = \gamma_{N,1}(\tau), \quad P_D = \gamma_{N,\frac{1}{n}}\left(\gamma_{N,1}^{-1}(P_{FA})\right), \qquad (4)$$

*where $\gamma_{x,y}$ is the lower incomplete gamma function with shape parameter $x$ and scale parameter $y$.*

Appendix C gives a sketch of the proof. The power of the test $P_D$ is an increasing function of $n$. Again, note that neither the token variables distribution nor the chunk length s affects the detectability.

Figure 3 illustrates the efficiency of our exploration strategy of the text space. Contrary to Sect. 3, we can easily reach a power close to 1 even at $P_{FA} = 10^{-6}$. For example, using $N = 9$ chunks of $n = 15$ drafts, assuming a medium-size text of $L = 512$ tokens and chunks of equal length, one only needs to generate $Nn = 135$ chunk drafts of $\ell = \lceil L/N \rceil = 57$ tokens to obtain a power of 0.96. This is equivalent to generating 15 texts of size 512. This is to compare to Sect. 3 where more than 3 million texts of 512 tokens are necessary to reach this power at $P_{FA} = 10^{-6}$.

## 5 Robust watermark detection

Until this point, our detector assumed that the text it receives is the generated text. Yet, before reaching the detector, a text might have been modified for legitimate or malicious reasons: it may be translated, curated, locally modified *etc*. This section assumes that the scores of individual tokens are distributed as standard Gaussian random variables to allow the derivation of closed-form solutions.

### 5.1 Robust detection

Detector (3) is neither optimal nor robust under attack because the insertion or the removal of tokens provokes a desynchronization: The detector no longer knows where each chunk starts and ends, which is necessary for computing the local $p$-value of individual chunks. More robustness comes with a global score simply summing up over all the tokens:

$$\Lambda_{\mathsf{rob}}(\mathbf{u}) = \sum_{i=1}^{N} \sum_{j=1}^{\ell} u_{(i-1)\ell+j} = \sum_{i=1}^{L} u_i. \qquad (5)$$

**Proposition 5.1.** *The robust detector has the following test performance when there is no attack:*

$$P_{FA} = \Phi(-\tau/\sqrt{L}), \quad P_D \approx \Phi\left(\frac{\Phi^{-1}(P_{FA}) + \sqrt{N}e(n)}{\sqrt{v(n)}}\right). \qquad (6)$$

*where $e(n)$ (resp. $v(n)$) is an increasing (resp. decreasing) quantity computed in Tab. 1.*

Appendix D gives a sketch of the proof. Figure 3 shows that the robust test (5) is slightly less powerful than the optimal test (3) when there is no attack. On the other hand, the next section shows that its power degrades smoothly under attacks contrary to (3).

### 5.2 A model of the robustness against attack

Despite the variety of modifications that can be performed on a watermarked text, all attacks have the same end effect: modifying the tokens and potentially the number of tokens. This amounts to modifying a proportion of scores to be distributed following $\mathcal{H}_0$ instead of $\mathcal{H}_1$ in the global score. Formally, the score $\Lambda_{\mathsf{rob}}(\mathbf{U})$ for a text of $L$ tokens becomes: (assuming $\alpha L$ is an integer)

$$\Lambda_{\mathsf{rob}}(\mathbf{U}) = \sum_{i=1}^{\alpha L} U_{\pi_0(i)} + \sum_{i=1}^{(1-\alpha)L} \bar{U}_{\pi_1(i)}, \qquad (7)$$

where $1 - \alpha$ is the proportion of scores impacted by the attack whose variables are denoted $\{\bar{U}_i\}$, and $\pi_0$ (resp. $\pi_1$) is mapping to the indices of the untouched scores (resp. modified tokens). It is important to note that it does not matter if the size of the attacked text differs from the generated text.

**Proposition 5.2.** *The robust detector has the following test performance under attack:*

$$P_{FA} = \Phi(-\tau/\sqrt{L}), \quad P_D \approx \Phi\left(\frac{\Phi^{-1}(P_{FA}) + \alpha\sqrt{N}e(n)}{\sqrt{1 + \alpha^2(v(n) - 1)}}\right). \tag{8}$$

Appendix E gives a sketch of the proof. In the end, the power of the test decreases smoothly with the strength $(1 - \alpha)$ of the attack. Once again, the power of the test does not depend on the text size.

# 6 Independence

This section discusses the assumptions made so far on the independence of the token and draft scores.

## 6.1 Token scores independence

All random variables $(U_i)$ associated with tokens must be independent and identically distributed within a text. This assumption has an impact on the power of the test and especially on the probability of false alarm. It must hold for non-watermarked text. Otherwise, the score of a chunk (or of a full text) is not properly normalized into a $p$-value.

We use the same technique as in [11] to enforce the token variable independence: the variable $U_i$ of the $i$-th token of a text depends not only on the secret key but also on the hashing of the current token and the $h - 1$ previous tokens in the text (a window of size $h$). Furthermore, this $h$-gram is appended to a list. If the $h$-gram is already stored in that list, the variable of the current token is discarded. By doing so, we ensure that *for a given text*, the $(U_i)$ are always different. Contrary to [11], this mechanism is enforced at the generation and the detection stages to ensure that both sides compute the same score.

## 6.2 Draft score independence

At the embedding, the scores of the $n$ drafts of a chunk are assumed to be independent. Yet, some correlations occur when parts of texts repeat across different drafts. For example, in a prompt asking "Who are you?", many drafts of the first chunk of the response likely start with "I am." This assumption only plays a role in the power of the detector, not its false-positive rate.

**Causal hashing window** At first sight, hashing the whole chunk draft for seeding each token variable brings draft score independence if the drafts differ at least by one token. Sadly, this creates a watermark that breaks even for a single modified token. *This forces us to use a causal window*: the score of each token depends only on itself and the previous tokens. The longer the hash window, the more likely a $h$-gram is new and the more diverse the scores. Yet, this diversity is obtained at the cost of robustness. Indeed, changing a single token in a text removes the watermark of $h$ tokens because their variables depend on the modification. Another weakness: at a given iteration, the $j$-th token ($j < h$) of every draft always refers to the same $h - j$ tokens since the previous chunk is identical for all these drafts. This means that the effective window size for this token is always smaller than $h$.

**Beam-searched enforced diversity** This weakness can be tackled by modifying the sampling procedure. It is important to guarantee a high diversity at the beginning of the chunk since this is where the effective window size is the smallest. We propose to generate the first $b$ tokens using a standard beam-search procedure, which deterministically returns $n$ different beginnings of the chunk. The rest of the draft is sampled normally. However, there is a trade-off. The smaller $b$, the more diversity between the chunks and the closer to the independence assumption we get, the more likely the beam-search procedure selects tokens in the tail of the distribution (for a large enough number of drafts $n$), the poorer the text quality.

## 6.3 Experimental validation

For the independence of the score variables, the experiment runs the detector on 100k Wikipedia entries from 2018 for ten different keys. Since the text of Wikipedia is free of any watermark, one should observe the $p$-values to be uniformly distributed, *i.e.* the empirical false alarm rate should

match the theoretical probability of false alarm. Figure 4a demonstrates that this assumption holds for any window size $h$. On the other hand, the robustness highly depends on the hashing window size, with $h = 6$ being a good trade-off between performance and robustness.

The second experiment measures how much we deviate from the draft score independence assumption. It generates 296 texts of 256 tokens on the three tasks from [29]. The entropy of the generated texts may be low for a given chunk so that some parts of a text may be redundant among the drafts. Figure 4b reports how much our scores under $\mathcal{H}_1$ deviate from the theoretical distribution.

This experiment demonstrates that the closer to 1 the number of tokens generated by beam search $b$, the closer the scores match the theoretical distribution. Notice that even for a large $b$, such as $b = 6$, there is a large improvement compared to a baseline WaterMax with $b = 0$. In Appendix I, we provide a more thorough experimental study of the parameters $h$ and $b$. In particular we show the trade-off between quality and detectability. In the rest of the paper, we select $b = 4$ where a small loss of quality is acceptable and $b = 6$ if a virtually lossless scheme is warranted.

## 7 Experiments

### 7.1 Experimental protocol

The evaluation is performed on the three long-form creative writing tasks of the '*Mark My Words*' benchmark [29]: news article generation, summarization of existing books, and writing of an invented story. This leads to the generation of 296 texts. We fix the maximum text size $L$ to 256 tokens for all tasks (see App. K for larger text sizes). The length of a chunk $\ell$ is fixed *a priori* to $L/N$ where $L$ is the maximum number of tokens allowed in the benchmark and $N$ the number of chunks.

All the evaluations in this section use the model Llama3-8b-Instruct [32, 3] (more models are tested in App. J). Its temperature $\theta$ varies to measure the impact of the text entropy on the watermark detectability. The watermarking scheme is not allowed to modify $\theta$ compared to the original LLM.

The evaluation of a watermarking scheme is based on three quantities: 1) the quality of the watermarked text relative to the non-watermarked text, 2) the detectability, and 3) the robustness.

For KGW and Aaronson's scheme, we use the implementation provided by [11] at `https://github.com/facebookresearch/three_bricks`. For the attack suite, we use the implementation of the '*Mark My Words*' benchmark found at `https://github.com/wagner-group/MarkMyWords`.

**Text quality** A number of metrics have been proposed to evaluate the quality of generated watermarked texts empirically. We tested the vast majority of these metrics: rating by LLM [29], similarity between BERT's embeddings [11], MAUVE [30], ROUGE [25] and perplexity measured by an oracle [21]. We arrived at the conclusion that perplexity and ROUGE-L are currently the only reliable

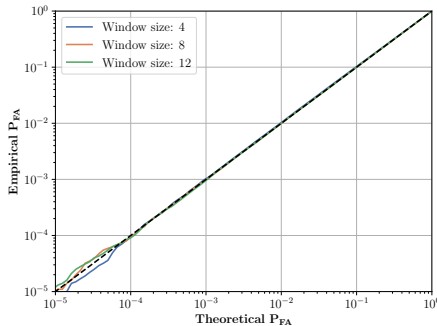

(a) Empirical *vs.* theoretical $P_{FA}$ for $\Lambda_{\mathrm{opt}}$, over 10×100k Wikipedia entries truncated at 256 tokens.

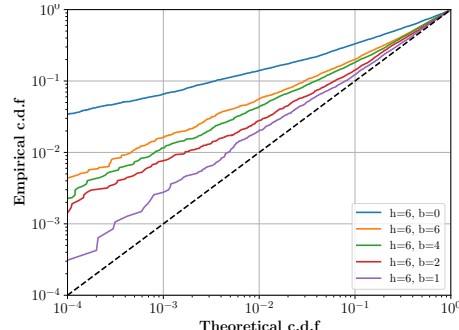

(b) Empirical *vs.* theoretical c.d.f of the score under $\mathcal{H}_1$, over 296×256 scores. Llama-2-7b-chat, Water-Max $(n, N) = (8, 16)$, $h = 6$.

Figure 4: Independence of token scores (a) and draft scores (b).

measures of watermarked text quality – in the sense that they are the only ones that consistently vary alongside watermark strength. This section reports the relative perplexity as measured by opt-2.7b as an oracle, computed as the average ratio between the perplexity of a watermarked text and the corresponding non-watermarked text. The text quality is evaluated by comparing the text generation with and without watermarking at the same LLM temperature $\theta$.

**Detectability**    The increasing use of LLM necessitates the use of small false-positive rates. In this section, we settled on reporting the true-positive rate at a conservative false-positive rate of $10^{-6}$. For a more complete picture, Appendix K also reports the median false-positive rate of each watermark algorithm at different text lengths, akin to the measure of "watermark size" recently proposed in [29].

**Robustness**    We use the attack suite of the '*Mark My Words*' (MMW) benchmark [29]. We report robustness by measuring the detectability $P_D$ of the watermark at $P_{FA} = 10^{-6}$ for each attack.

Error bars are reported using one standard error for perplexity, using the standard methodology, and detectability, computed using 5000 bootstrap samples for power at $P_{FA} = 10^{-6}$.

## 7.2  State-of-the-art methods

The recent MMW benchmark considers four techniques names as 'binary' [8], KGW [21], Aaronson [1] and 'inverse transform' [23]. It concludes that today's best methods are KGW and Aaronson, we thus restrict our comparisons to them. Except when specified otherwise, the watermark schemes all use a window size of $h = 6$ for hashing.

A fair comparison considers watermarking schemes at an equivalent level of text quality. Aaronson's scheme is theoretically distortion-free and experimentally almost lossless with respect to empirical relative perplexity and the ROUGE-L score. We should ideally tune WaterMax and KGW such that they are also almost lossless. This is not a problem for WaterMax as long as the number of tokens generated by a beam-search $b$ is small enough. We choose the setting $(N, n) = (16, 10)$, which provides a good compromise between robustness and complexity. As for KGW [21], we fix $\delta = 2.0$ for an acceptable loss of quality, or $\delta = 3.0$ for a tangible loss of quality. This gives a clear advantage to KGW. We set its green-list ratio $\gamma = 0.5$ following the recommendation of Piet et al. [29].

## 7.3  Results

**Quality vs detectability**    Figure 5 summarizes the benchmark outcomes. They confirm that Water-Max achieves a detectability close to the theoretical prediction – in our case close to 1 at $P_{FA} = 10^{-6}$ – while, at the same time, incurring a minimal loss in text quality. In particular, for a number of tokens generated by beam-search $b = 6$, there is virtually no observed loss in terms of perplexity. Furthermore, its performance is independent of the temperature of the LLM, demonstrating the efficient use of the entropy by working on chunks instead of individual tokens. On the other hand, Aaronson's scheme achieves high detectability only for high temperatures (seldom used in practice), whereas KGW achieves high detectability at the price of a text quality loss: at $\delta = 3.0$ despite a relative perplexity significantly larger than WaterMax, KGW is still far less detectable.

**Robustness**    Figure 6 summarizes the results for the attack suite of the MMW benchmark. The watermark parameters are fixed to ensure the most negligible quality loss possible: $b = 6$ for WaterMax and $\delta = 2.0$ for KGW. Without many surprises, no algorithm can resist the powerful re-translation attacks at this $P_{FA}$ regime since most words, as well as their order, are modified. The 'typos' attacks can safely be disregarded as they modify every token and make the text barely legible in practice. This leaves the attacks that modify only parts of the text. For low temperatures, WaterMax is by far the most robust. Notably, the robust detector (5) can resist far more attacks, with significantly higher detectability against 'synonym', 'misspelling' and 'swap' attacks.

Interestingly, for temperatures starting at $1.0$, Aaronson's scheme is more robust than WaterMax for specific attacks. This is explained by the high reliance of this method on entropy as we show in Appendix J as well as the high variance of its $p$-values. Given enough entropy, some portions of the watermarked text present unusually low $p$-values, statistically leading to more robustness. On the other hand, WaterMax produces texts with $p$-values that are more concentrated around their mean.

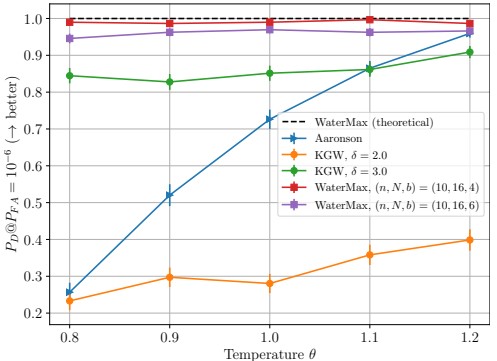
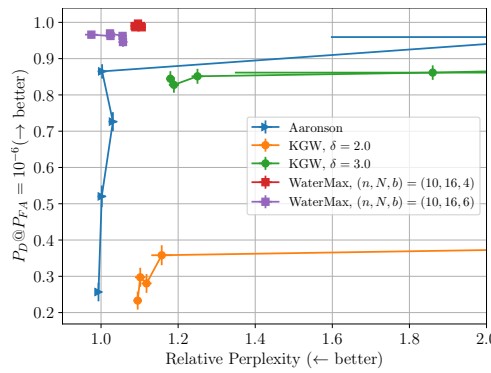

(a) Detectability as a function of the LLM temperature $\theta$. WaterMax is the only scheme for which detectability is maximal for any $\theta$.

(b) Detectability as a function of text quality. Each point corresponds to a temperature from the figure on the left.

Figure 5: Detectability against quality of watermarking schemes using Llama-3-8b-Instruct with nucleus sampling ($top_p = 0.95$) and hashing window $h = 6$.

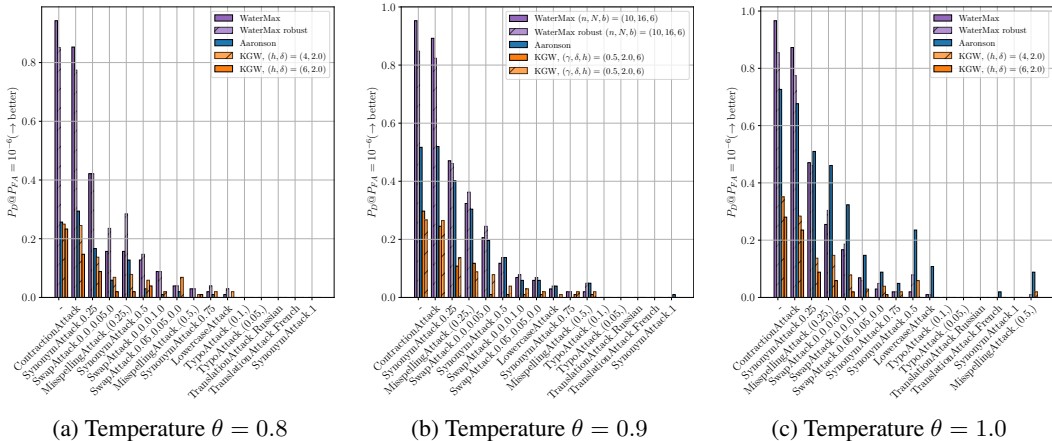

(a) Temperature $\theta = 0.8$      (b) Temperature $\theta = 0.9$      (c) Temperature $\theta = 1.0$

Figure 6: Robustness against the attacks of MMW benchmark. Llama-3-8b-Instruct with nucleus sampling ($top_p = 0.95$). WaterMax parameters $(N, n, b) = (16, 10, 6)$. The hashing window size is fixed to $h = 6$ for all schemes.

## 7.4 Computational complexity

At face value, the computational cost of our scheme is nothing more than $Nn$ text generations, the computation of the scores being negligible. However, the cost of adding one more chunk is higher than the cost of generating one more draft. Indeed, the $n$ drafts of a chunk are sampled independently (or by a beam search), making their computation highly parallelizable on modern GPUs. On the other hand, we cannot parallelize the computations over the chunk. Since $N$ has the highest impact on the power of WaterMax, it is the main limiting factor for its performance.

See Figure 7 for experimental running times. As a rule of thumb, we advise to use parameters fixed as $(n, N, b) = (10, 16, 6)$ as good trade-off between computational complexity and detectability. This still incurs a high cost compared to classic watermarking schemes. Under these settings, a text generated with WaterMax takes on average five times longer to generate than KGW or Aaronson's scheme.

However, despite a higher total computational cost, the latency can be kept relatively low since the text is generated chunk by chunk. Each chunk can be delivered to the user gradually (e.g. using a buffer) in order to make the method practical in a real-life setting. Note that more chunks lead to lower latency **and** better detectability with a negligible cost to text quality. We report, under the same settings, the generation time for one chunk as a measure of latency in Figure 7. In particular, we

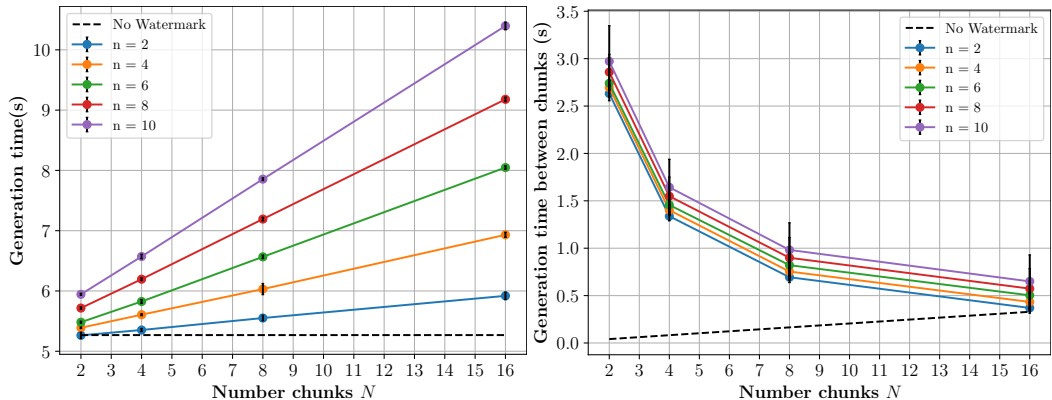

Figure 7: (left) Total generation time in seconds of WaterMax as a function of the number of chunks $N$ and the number of drafts/chunk $n$ for texts of 256 tokens using sampling on a Nvidia A100 with **Llama-2-chat-hf**. (right) Generation latency defined as the average generation time between two chunks compared to the generation time of the same number of tokens with a baseline LLM. Note that the baseline time between two tokens is 0.021s.

show that under the advised parameters, the latency of WaterMax is only twice as long as that of the original LLM.

# 8 Conclusion

Contrary to previous art, the design of WaterMax starts from a detector and then constructs the generator to maximize the detection power. As a result, our watermark scheme offers a host of compelling benefits. It achieves high quality and robustness even on short texts, and importantly, it does not modify any component of the LLM, preserving its integrity and functionality. This design also complies with common add-ons in the literature, like adapting the sampling temperature [19] or embedding short messages [11].

These advantages come at the cost of computational complexity, which stays limited thanks to the exploration strategy and the parallelization possibilities of modern GPUs. Another idea left for future work is the distillation of WaterMax, a process that involves fine-tuning an LLM to natively produce watermarked text. This would definitively get rid of the only drawback of WaterMax.

# Acknowledgment

Work supported by French ANR / AID under the chaire SAIDA (ANR-20-CHIA- 0011-01). Experiments presented in this paper were carried out using the Grid'5000 testbed, supported by a scientific interest group hosted by Inria and including CNRS, RENATER and several Universities as well as other organizations (see `https://www.grid5000.fr`).

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

# A   Possible detection schemes

## A.1   Baseline schemes

This section describes the detection of the three main LLM watermarking schemes. They share the same process:

- The received text is decomposed into a sequence of, say $L$, tokens
- A sliding window of size $h$ forwards the selected tokens to a cryptographic function which computes a hash that is xored with the secret key to compose the seed of a Pseudo Random Number Generator
- The variable $u_i$ is computed for the $i$-th token, $i > h$ based on the output of the PRNG seeded by the $h$ previous tokens
- The score is the sum of the variable: $s = \sum_{i=h+1}^{L} u_i$
- The score is converted into a $p$-value: $p = 1 - F_S(s; L)$.

The watermark embedding methods are not detailed because our scheme does not use them. This summary is inspired by the work of Fernandez et al. [11].

For the detection of [22], the output of the PRNG is used to create a 'green' list which is a subset of $\mathcal{V}$ of size $\gamma|\mathcal{V}|$. Variable $U_i$ is set to 1 if the $i$-th token belongs to this list, otherwise $U_i = 0$. Therefore, under $\mathcal{H}_0$, $U_i \overset{\text{iid}}{\sim} \mathcal{B}(\gamma)$ and $S$ follows a binomial distribution $\mathcal{B}(L - h, \gamma)$. This makes

$$p = I_\gamma(s, L - h - s + 1), \tag{9}$$

where $I$ is the regularized incomplete Beta function.

For the detection of [1], the output of the PRNG is used to draw $U_i \overset{\text{iid}}{\sim} \mathcal{E}(1)$ so that $S$ is Gamma distributed under $\mathcal{H}_0$. This makes:

$$p = \frac{\Gamma(L - h, s)}{\Gamma(L - h)}, \tag{10}$$

where $\Gamma$ is the upper incomplete gamma function. This formulation also holds for [8] with the reduction to binary alphabet $|\mathcal{V}| = 2$.

For the detection of [23], the output of the PRNG is used to draw $U_i$ whose distribution matches the distribution of the product of two independent r.v. $\mathcal{U}([-1/2, 1/2])$, *i.e.* $F_U(u) = 2u(1 - \log(4|u|)) + 1/2$. There is no closed form to turn the global score into a $p$-value. The authors use either a bound (see their Lemma 2.4) or an empirical $p$-value computed by a costly Monte Carlo simulation with random seeds. This is the reason why this studies [23] fixes the probability of false alarm to a relatively large value, *i.e.* $10^{-2}$.

Since Sect. 4.4 shows that the distribution of $U_i$ has no impact, we opt for the simple choice $U_i \overset{\text{iid}}{\sim} \mathcal{N}(0; 1)$ and the $p$-value reads as:

$$p = \Phi\left(-\frac{s}{\sqrt{L - h}}\right). \tag{11}$$

## A.2   Variants

Numerous works have proposed improvements of the KGW and Aaronson's scheme. We herein describe some relevant works and their applicability to the WaterMax framework.

**KGW Variants**   Improvements to KGW can be broadly categorized into two classes: modification to the sampling mechanism and improvements to the bias assignment. Some recent works modifying the sampling mechanism are: [36] which propose to complement baseline KGW with contrastive search to improve text quality as well as inserting a second watermark, [18] uses a multi-objective optimization procedure to conserve the semantic coherence of the watermarked text Works on better bias assignments include: [15] which propose to cluster tokens which are semantically related and assign similar biases to them in order to resist re-translation attacks, [7] builds the green and red

list using a similar clustering method, [26] adapt the bias depending on the entropy of the current completions in order to improve detectability and text quality. Note that any watermarking methods relying on modifying the biases could directly be used by WaterMax as a score function as long as a $p$-value can be computed from them. Similarly any modified sampling method could also be applied as long as they can output different texts from the same prompt.

**Aaronson's scheme variants**    Most work dealing with improving on the Gumbel-trick of Aaronson's scheme are not explicitly variants but other algorithms with the distortion-free property, or methods to improve on the ad-hoc detector of the original work. Noteworthy distortion-free algorithms include the 'binary' [8] 'inverse transform' [23] methods, the latter which actually includes a variant of Aaronson. The work in [10] is notable for being an asymmetric (somewhat fragile) watermark.Though technically presented as a variant of KGW, one of the proposed method in [16] can be interpreted as variant of Aaronson working on group of tokens. Finally, [24] proposes an improved score function for Aaronson through the design of a general statistical framework.

**Extensions to multi-bit watermarking**    Even though this work is solely concerned with zero-bit watermarks, we note that some works have proposed mechanisms to extend current text watermark methods to multi-bit versions, allowing for example the identification of specific users. Of note, [11] have proposed a general mechanism applicable to every current zero-bit method by associating a key to every possible message and designing an efficient detector through a circular shift of the keys, [31] extends KGW to multi-bit using BCH and error-correcting codes.

**Watermark distillation**    To this we add an important work from Gu et al. [13], especially relevant for WaterMax, showing the possibility to directly train a LLM to generate watermarked text using either watermarked samples or directly the token distribution. Since the only real cost of WaterMax is the computational complexity of sampling chunk of texts, distilling WaterMax would allow to bypass the sampling cost entirely.

# B   Proof of proposition 3.1

The main ingredient of our scheme is a watermark detection that outputs a $p$-value for any text input, whatever its length. This is the case for the main works in the literature, like [1, 21, 8] and to some extent [23] (see App. A). Of note,

- In theory, for a continuous score $S$, the $p$-value $P$ computed from a random non-watermarked text is uniformly distributed over $[0, 1]$ with p.d.f. $f(p|\mathcal{H}_0) = \mathbb{1}_{[0,1]}(p)$.

- In practice, the computation of the $p$-value can be hazardous on texts with repetitions of token sub-sequences and Fernandez et al. [11] indicate how to fix this issue.

For a given prompt, the generator creates $n$ independent texts, each with possibly a different number of tokens, computes their $p$-values $P_j \overset{\text{iid}}{\sim} \mathcal{U}_{[0,1]}$, and outputs the text with the minimum $p$-value. Note that in order for the $p$-value to follow a uniform distribution under $\mathcal{H}_0$, we assume the c.d.f of the scores to be continuous. A well-known results in statistics states that the minimum of $n$ independent $p$-values follow a beta distribution $B(1, n)$ whose p.d.f. is $f(p|\mathcal{H}_1) = n(1-p)^{n-1}$.

The detector receives an unknown text and decides between two hypotheses:

- $\mathcal{H}_0$: The text is not watermarked. The $p$-value has not been 'minimized' and follows the uniform distribution.

- $\mathcal{H}_1$: The text is watermarked. Assuming no attack was performed, the $p$-value is distributed as a minimum of $n$ independent uniform variables. We assume the detector knows the value of $n$.

Since the distributions under both hypotheses are known, the detector performs a simple test, and the Neyman-Pearson lemma states that the most powerful test under a given probability of false-alarm $P_{FA}$ is the (log) likelihood-ratio test:

$$\Lambda(p) \triangleq \log\left(\frac{f(p|\mathcal{H}_1)}{f(p|\mathcal{H}_0)}\right) \underset{\mathcal{H}_1}{\overset{\mathcal{H}_0}{\lessgtr}} \tau, \tag{12}$$

where $\tau$ is a threshold which depends on $P_{FA}$. In our setting, $\Lambda(p) = (n-1)\log(1-p) + \log(n)$, which is a monotonic function of $p$. In other words, comparing $p \lessgtr_{\mathcal{H}_0}^{\mathcal{H}_1} P_{FA}$ is equivalent to the Neyman-Pearson test. The power of the test is defined as the probability of detecting a watermarked text and turns out to be an increasing function w.r.t. $n$:

$$P_D \triangleq \mathbb{P}(P < P_{FA}|\mathcal{H}_1) = \int_0^{P_{FA}} f(p|\mathcal{H}_1)dp = 1 - (1 - P_{FA})^n. \tag{13}$$

## C   Proof of proposition 4.1

The optimal detector (3) aggregates a vector $\mathbf{p}$ of $N$ observed local $p$-values:

$$\Lambda_{\text{opt}}(\mathbf{p}) = -\sum_{i=1}^{N} \log(1 - p_i) \lessgtr_{\mathcal{H}_1}^{\mathcal{H}_0} \tau. \tag{14}$$

The received text is composed of chunks whose local $p$-values are distributed as beta distributions $B(1,1)$ (*i.e.* $\mathcal{U}_{[0,1]}$) under $\mathcal{H}_0$ or $B(1,n)$ under $\mathcal{H}_1$ when there is no attack (see Sect. B). Knowing that if $X \sim B(1,\beta)$ then $-\log(1-X) \sim \mathcal{E}(\beta)$, $\Lambda(\mathbf{P})$ is the sum of independent exponential r.v. and thus follows a Gamma distribution under both hypothesis but with different parameters:

$$\begin{cases} \Lambda_{\text{opt}}(\mathbf{P}) \sim \Gamma(N, 1) & \text{under } \mathcal{H}_0, \\ \Lambda_{\text{opt}}(\mathbf{P}) \sim \Gamma\left(N, \frac{1}{n}\right) & \text{under } \mathcal{H}_1. \end{cases} \tag{15}$$

This leads to the characterization of the test's performance:

$$P_{FA} = \gamma_{N,1}(\tau), \tag{16}$$

$$P_D = \gamma_{N,\frac{1}{n}}\left(\gamma_{N,1}^{-1}(P_{FA})\right). \tag{17}$$

where $\gamma_{x,y}$ is the lower incomplete gamma function with shape parameter $x$ and scale parameter $y$.

## D   Proof of proposition 5.1

Assume the received text is composed of $N$ chunks of $\ell$ tokens each so that its total number of tokens $L = N\ell$. Under $\mathcal{H}_0$, the token variables are independent and normal distributed so that $\Lambda_{\text{rob}}(\mathbf{U}) \sim \mathcal{N}(0; L)$. It is thus easy to find the threshold $\tau$ ensuring a probability of false alarm $P_{FA}$: $\tau = -\sqrt{L}\Phi^{-1}(P_{FA})$.

Under $\mathcal{H}_1$, $\Lambda_{\text{rob}}(\mathbf{U})$ can be written as the sum of $N$ independent chunk scores: $\Lambda_{\text{rob}}(\mathbf{U}) = \sum_{i=1}^{N} M_i^{(n)}$. These chunk scores are distributed as the maximum of $n$ Gaussian variables $\mathcal{N}(0; \ell)$, corresponding to the score of the drafts. Those maxima have the c.d.f. $F_{M^{(n)}}(x) = \Phi\left(x/\sqrt{\ell}\right)^n$ and one can compute numerically the expectation $\mathbb{E}(M^{(n)}) = \sqrt{\ell}e(n)$ and the variance $\mathbb{V}(M^{(n)}) = \ell v(n)$, see Tab. 1. As far as we know, there is no closed-form for these quantities, but classical results in concentration inequalities for the maximum show that $e(n)$ scales as $\sqrt{\log(n)}$ and $v(n)$ as $1/\log(n)$ [4].

An approximation for $N$ large enough is $\Lambda_{\text{rob}}(\mathbf{U}) \sim \mathcal{N}(\mu_1; \sigma_1^2)$ with

$$\mu_1 = N\sqrt{\ell}e(n) = \sqrt{NL}e(n), \tag{18}$$

$$\sigma_1^2 = N\ell v(n) = Lv(n). \tag{19}$$

This gives the following power of the test:

$$P_D = \Phi\left(\frac{\Phi^{-1}(P_{FA}) + \sqrt{N}e(n)}{\sqrt{v(n)}}\right). \tag{20}$$

Table 1: Expectation and variance of the maximum of $n$ independent normal random variables.

| $n$ | 1 | 2 | 3 | 4 | 5 | 6 | 7 | 8 | 9 | 10 |
|---|---|---|---|---|---|---|---|---|---|---|
| $e(n)$ | 0 | 0.56 | 0.84 | 1.03 | 1.16 | 1.26 | 1.35 | 1.42 | 1.48 | 1.54 |
| $v(n)$ | 1 | 0.68 | 0.56 | 0.49 | 0.45 | 0.42 | 0.40 | 0.37 | 0.36 | 0.34 |

# E   Proof of proposition 5.2

This section derives the distribution of *variables of individual tokens* under $\mathcal{H}_1$.

We first express the distribution of a vector of $L$ i.i.d. normal variables, given that their sum equals a given value $S$:

$$\left(\mathbf{U} \mid \sum_{i=0}^{L} U_i = s\right) \sim \mathcal{N}\left(s\mathbf{1}_L/L, \mathbf{I}_L - \mathbf{1}_L\mathbf{1}_L^\top/L\right) \tag{21}$$

with $\mathbf{1}_L$ the vector of all ones in $\mathbb{R}^L$ and $\mathbf{I}_L$ the identity matrix of dimension $L \times L$.

The attacks leaves $\alpha L$ variables untouched (assuming $0 \leq \alpha \leq 1$ and $\alpha L \in \mathbb{N}$). This is encoded in a matrix $\mathbf{A} \in \{0,1\}^{\alpha L \times L}$ which selects these variables. This means that $\mathbf{A}$ has exactly one single entry equal to 1 in each row, $\mathbf{A}\mathbf{A}^\top = \mathbf{I}_{\alpha L}$, and $\mathbf{A}\mathbf{1}_L = \mathbf{1}_{\alpha L}$.

The sum of these $\alpha L$ variables can be written as $\mathbf{1}_{\alpha L}^\top \mathbf{A}\mathbf{U}$. This makes:

$$\left(\mathbf{1}_{\alpha L}^\top \mathbf{A}\mathbf{U} \mid \sum_{i=0}^{L} U_i = s\right) \sim \mathcal{N}\left(\alpha s; L\alpha(1-\alpha)\right). \tag{22}$$

This distribution is independent of $\mathbf{A}$: It does not matter which tokens are left unchanged. For $\alpha = 1$, there is no attack and the variance is null because $\mathbf{1}_{\alpha L}^\top \mathbf{A}\mathbf{U} = \mathbf{1}_L^\top \mathbf{U} = s$. For $\alpha = 0$, the attack has modified all the token variables and the variance is null because $\mathbf{1}_{\alpha L}^\top \mathbf{A}\mathbf{U} = 0$.

According to App. D, an approximation for $N$ large enough is $S \sim \mathcal{N}(\mu_1; \sigma_1^2)$. The law of total variance gives

$$\mathbf{1}_{\alpha L}^\top \mathbf{A}\mathbf{U} \sim \mathcal{N}\left(\alpha\mu_1; L\alpha(1-\alpha) + \alpha^2\sigma_1^2\right). \tag{23}$$

On top of this, the attack adds $(1-\alpha)L$ tokens distributed under $\mathcal{H}_0$, which amounts to add a noise distributed as $\mathcal{N}(0, (1-\alpha)L)$ to the above score. In the end, the global score is distributed as

$$\Lambda_{\mathsf{rob}}(\mathbf{U}) \sim \mathcal{N}\left(\alpha\sqrt{NL}e(n); L(1 + \alpha^2(v(n) - 1))\right). \tag{24}$$

We thus obtain an approximate expression of the power of the robust test:

$$P_D = \Phi\left(\frac{\Phi^{-1}(P_{FA}) + \alpha\sqrt{N}e(n)}{\sqrt{1 + \alpha^2(v(n) - 1)}}\right). \tag{25}$$

Observe that, once again, the power of the test does not depend on the size of the text. More precisely, it does depend indirectly on it given that $\alpha$ can only take a finite number of values, the number of which decreases with the size of the text. For example, if the text is composed of $L$ tokens, $\alpha \in \{0, 1/L, \ldots, 1 - 1/L, 1\}$.

Note that this analysis is general in the sense that it does not depend on the existence of an attack at a given $\alpha$ resulting in an acceptable quality. We believe the role of the watermark designer is to ensure the highest possible detectability whatever the strength of the attack; the design of attacks preserving text quality is the burden of the attacker. An analysis of the robustness of a watermark should thus be **comprehensive**, providing the performance of the watermark under any possible type of attacks, existing or not, and **independent of text quality**.

# F Pseudo-code of the watermark embedding

---

**Algorithm 1** Iterative watermarking of generated texts

---

**Input:** prompt pr, parameters $(N, n, m, \ell)$, $p$-value computation Detect, LLM Generate
**Initialize:** $\mathbf{x}_j = [], \forall 1 \leq j \leq m$
**for** $i = 1$ **to** $N$ **do**
    **for** $j = 1$ **to** $m$ **do**
        context $= [\text{pr}\|\mathbf{x}_j]$
        **for** $k = 1$ **to** $n$ **do**
            $\nu = n(j-1) + k$
            $\mathbf{y}_\nu = \text{Generate}(\text{context}, \ell)$
            $p_\nu = \text{Detect}([\mathbf{x}_j\|\mathbf{y}_\nu])$
        **end for**
    **end for**
    $(\nu_1, \cdots, \nu_m) = \arg\min_{1 \leq \nu \leq mn} p_\nu$
    **for** $j = 1$ **to** $m$ **do**
        $\mathbf{x}_j = [\mathbf{x}_{\lfloor \nu_j/n \rfloor + 1}\|\mathbf{y}_{\nu_j}]$
    **end for**
**end for**
$l = \arg\min_{1 \leq j \leq m} \text{Detect}(\mathbf{x}_j)$
**Output:** $\mathbf{x}_l$

---

# G Experimental running time

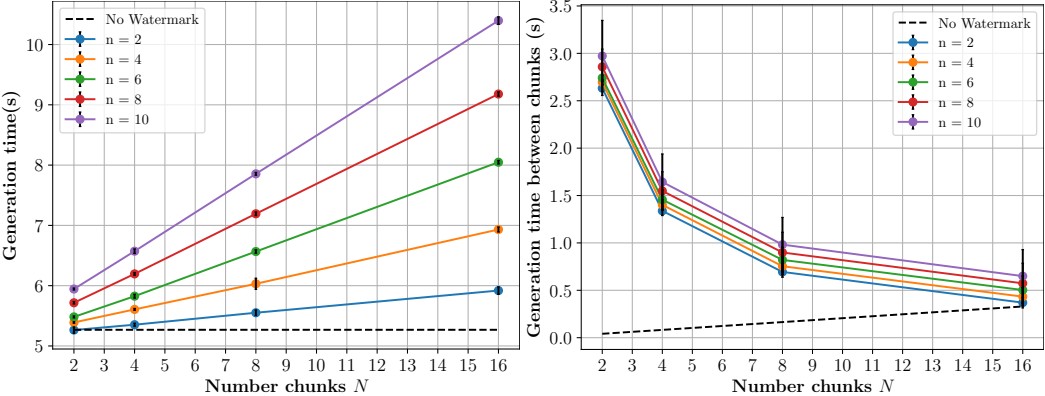

Figure 8: (left) Total generation time in seconds of WaterMax as a function of the number of chunks $N$ and the number of drafts/chunk $n$ for texts of 256 tokens using sampling on a Nvidia A100. (right) Generation latency defined as the average generation time between two chunks compared to the generation time of the same number of tokens with a baseline LLM. Note that the baseline time between two tokens is 0.021s.

We report the running time averaged on 5 texts of 256 tokens generated using *Llama-2-chat-hf* at temperature 1.0 and nucleus sampling ($top_p = 0.95$) for different parameters in Figure 8.

Overall, generating the 296 texts of the 'Mark My Words' benchmark on a single A100, with maximum batch-size, takes approximately 6min for Aaronson and KGW. For WaterMax at $(n, N, b) = (10, 16, 6)$, it takes approximately 30min. The resource usage report of the cluster used to run the experiments, taking into account both CPU and GPU cores, indicates a total use of 5024 core hours.

# H  WaterMax is almost distortion-free

This appendix elaborates a theoretical model to justify why WaterMax is close to distortion-freeness.

Consider a given iteration of WaterMax where one particular draft among $n$ is selected as the output chunk. Denote the set of possible chunks by $\mathcal{C}$. If the chunks are all composed of $\ell$ tokens, then $|\mathcal{C}| = |\mathcal{V}|^\ell$. Each chunk $c_i \in \mathcal{C}$ has a probability $p_i$ to be sampled by the LLM. We denote by $\tilde{p}_i$ the probability that WaterMax selects this chunk. This probability is computed on expectation over the set of secret keys.

A watermarking scheme is distortion-free if $\tilde{p}_i = p_i$, $\forall p_i \in [0, 1]$. This appendix shows this is not the case for WaterMax. Yet, since we operate on chunks, *i.e.* sequences of tokens, and not individual tokens, their probabilities are small in practice. In this regime, we show that $\tilde{p}_i \approx p_i$, which theoretically justifies the high text quality.

## H.1  $n$ draws with replacement

This version of WaterMax samples $n$ drafts in $\mathcal{C}$. This corresponds to draws with replacement because a draft can be sampled several times.

Denote by $X_i$ the number of times the $i - th$ chunk is drawn over $n$ samples. It follows the binomial distribution $X_i \sim \mathcal{B}(n, p_i)$, $\forall i \in \{1, \ldots, |\mathcal{C}|\}$.

Denote by $Y_i$ the binary random variable s.t. $Y_i = 1$ if $X_i > 0$. It follows the Bernoulli distribution $Y_i \sim \mathcal{B}(1 - (1 - p_i)^n)$, $\forall i \in \{1, \ldots, |\mathcal{C}|\}$.

Denote by $\mathcal{S}$ the subset of chunks which have been sampled, *i.e.* $\mathcal{S} = \{c_i : Y_i = 1\} \subset \mathcal{C}$. The size of this set is a random variable ranging from 1 to $n$ and whose expectation is $\mathbb{E}(|\mathcal{S}|) = \sum_{i=1}^{|\mathcal{C}|} Y_i = |\mathcal{C}| - \sum_{i=1}^{|\mathcal{C}|} (1 - p_i)^n$.

For a given watermark secret key, *i.e.* a given instance of the token variables $\{U_j\}_{j=1}^{|\mathcal{V}|}$, WaterMax computes the score of each draft in $\mathcal{S}$, and it chooses the one with the maximum score. Over the ensemble of secret keys, there is no reason why a given draft is more likely to be chosen due to the symmetry of the random token variables. Therefore, a draft in a given subset $\mathcal{S}$ has a probability $1/|\mathcal{S}|$ to be chosen on expectation over the secret keys. This yields the following global probability:

$$\tilde{p}_i = \sum_{\mathcal{S}:c_i \in \mathcal{S}} \frac{1}{|\mathcal{S}|} \mathbb{P}(\mathcal{S}). \tag{26}$$

**Upper bound**  The above summation includes the subset $\mathcal{S} = \{c_i\}$ which happens with probability $p_i^n$. For all the other subsets, which include $c_i$, $|\mathcal{S}| \geq 2$. This gives an upper bound:

$$\tilde{p}_i \leq UB(p_i) = p_i^n + \frac{1}{2}\left(1 - (1 - p_i)^n - p_i^n\right) = \frac{1}{2}\left(1 - (1 - p_i)^n + p_i^n\right). \tag{27}$$

**Lower bound**  The above summation can be written as $\tilde{p}_i = \sum_{k=1}^{n} \sum_{\mathcal{S}:X_i=k} \frac{1}{|\mathcal{S}|} \mathbb{P}(\mathcal{S})$. When $X_i = k$, there are at most $n - k$ other unique sampled chunks: $|\mathcal{S}| \leq n - k + 1$. This makes the following lower bound:

$$\tilde{p}_i \geq LB(p_i) = \sum_{k=1}^{n} \frac{1}{n - k + 1}\binom{n}{k} p_i^k (1 - p_i)^{n-k} = \frac{1 - (1 - p_i)^{n+1} - p_i^{n+1}}{(n + 1)(1 - p_i)}. \tag{28}$$

**Remarks**  Note that $\tilde{p}_i = p_i$, $\forall p_i \in [0, 1]$ when $n \in \{1, 2\}$ because the lower and upper bounds correspond to the identity. The choice $n = 1$ is not an option since it corresponds to the regular use of the original LLM without watermarking. The choice $n = 2$ allows watermarking with the distortion-free property. The price to pay is a low robustness.

For $n > 2$, WaterMax is not distortion-free. Indeed the upper bound (27) is lower than $p_i$ for $p_i > 1/2$ showing that $\tilde{p}_i$ does not equal $p_i$ on the interval $(1/2, 1)$. However, note that

$$\lim_{p_i \to 0} \frac{LB(p_i)}{p_i} = 1. \tag{29}$$

This property explains why WaterMax reaches text quality as good as other distortion-free watermarking schemes. WaterMax works with chunks of text whose probabilities are all very small, contrary to other schemes working on tokens which sometimes have a picky distribution (one token gets almost all the probability mass). Therefore, missing the distortion-free property for large $p_i$ is irrelevant because this case never happens in practice when dealing with chunks. On the contrary, having $\tilde{p}_i \approx p_i$ locally for $p_i \to 0$ is key.

### H.2 $n$ draws without replacement

This version of WaterMax samples $n$ drafts in $\mathcal{C}$ making sure that they are all different thanks to the beam-search suggested in Sec. 6.2. This corresponds to draws without replacement because a draft cannot be sampled twice. In other words, $|\mathcal{S}| = n$. Equation (26) simplifies to:

$$\tilde{p}_i = \frac{1}{n}\mathbb{P}(c_i \in \mathcal{S}). \tag{30}$$

However, the probability $\mathbb{P}(c_i \in \mathcal{S})$ is cumbersome to calculate as it relates to the multinomial Wallenius noncentral hypergeometric distribution. When all the chunk probabilities are very small, one can show that [12]:

$$\tilde{p}_i \approx \frac{p_i}{n}\sum_{k=0}^{n-1}(1-p_i)^k = \frac{1-(1-p_i)^n}{n} \overset{p_i \to 0}{\longrightarrow} p_i. \tag{31}$$

The same rationale as above applies: WaterMax is not distortion-free but tends to this property when all the chunks have small probabilities. This is what happens in practice when working with long chunks.

## I  Impact of hashing window size and beam-search tokens

Section 6 discusses how the window hashing size $h$ as well the number of tokens generated by beam-search $b$ can help to increase detectability. Both parameters, however, lead to a trade-off. First, a higher $h$ provides better detectability when no attack is performed on the text but makes the watermark less robust since modifying one token leads to at most $h$ scores, which become unusable. Secondly, a lower $b > 0$ leads to better detectability by enforcing diversity between chunk drafts, at no cost to robustness, but at the cost of text quality since the beam-search has more chance of selecting tokens in the tail of the distribution. Note that $b = 0$ corresponds to the baseline WaterMax, which is empirically lossless.

To select the best trade-off, we generate 296 texts of 256 tokens following the same protocol described in Sect. 6: we use Llama-3-8b-Instruct with temperature $\theta = 1.0$, nucleus sampling ($top_p = 0.95$), and WaterMax at $(n, N) = (8, 16)$. We report the results in Fig. 9-11.

From Figure 9-10, we see that the most significant improvement in the trade-off detectability vs. quality is by going from $b = 0$ to $b = 6$. This is done at almost no cost to the text quality. A larger gain in detectability is obtained for $1 \leq b < 6$ but with a significant loss of quality.

Regarding the window size, $h = 2$ leads to a weak scheme. An acceptable performance, close to $P_D = 1.0$ at $P_{FA} = 10^{-6}$ without attack, requires $h \geq 4$. However, Figure 11 reveals that a close match to the theoretical score distribution under $\mathcal{H}_1$ is only reached for $h = 6$.

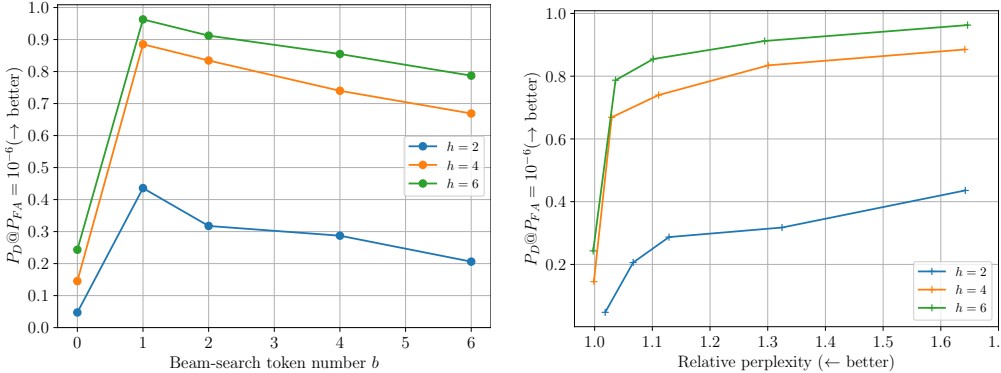

Figure 9: Detectability as a function of the number of tokens generated by beam-search $b$

Figure 10: Detectability as a function of the text quality. The points correspond to the different $b$ used in Fig. 9

.

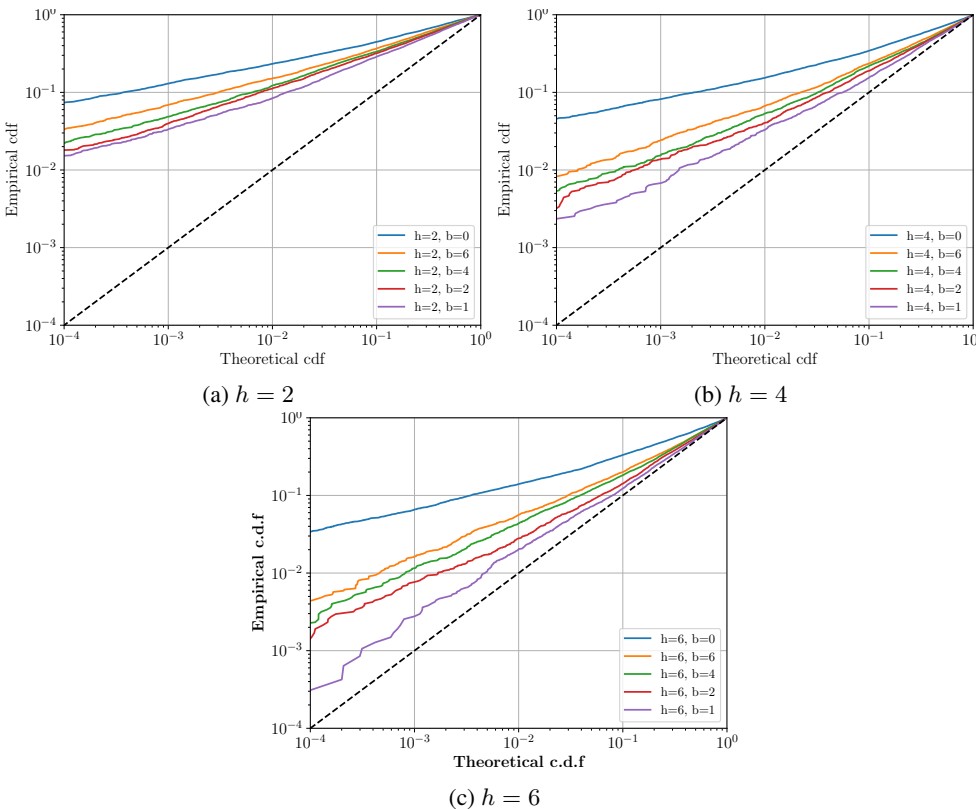

(a) $h = 2$

(b) $h = 4$

(c) $h = 6$

Figure 11: Score independence as a function of the window hashing size $h$ and number of tokens generated by beam-search $b$.

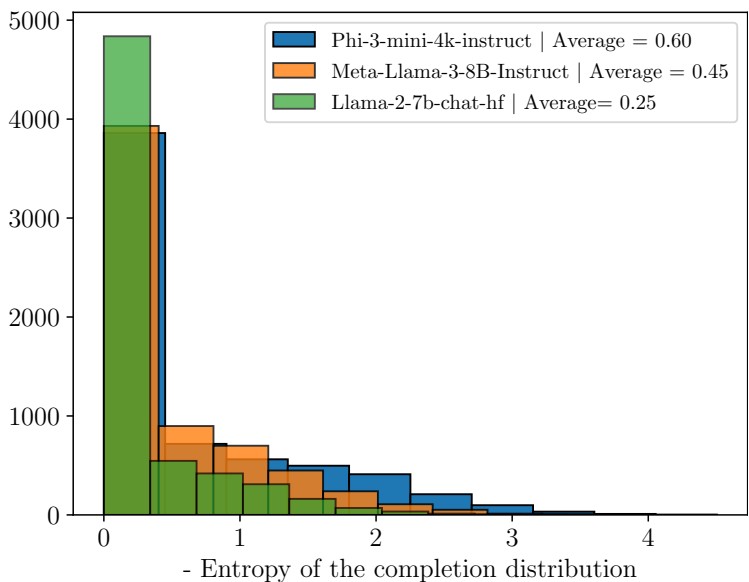

Figure 12: Entropy of the distribution of $64 \times 100$ completions for different LLM at temperature $\theta = 1.0$ using nucleus sampling ($top_p = 0.95$).

## J  Impact of the LLM and its entropy

This appendix studies the impact of the LLM on the watermark performance. As we alluded to numerous times, the performance of the watermark of KGW and Aaronson is highly dependent on the entropy of the completion of individual tokens. This dependence is well-illustrated by the impact of temperature on their performance. However, this also means that we can expect vastly different behaviors of these schemes depending on the LLM used since, depending on how the LLM was trained and fine-tuned, the average entropy of the completions varies wildly.

To demonstrate this point, we repeated the experiments in Sect. 7 for two other models: the older *Llama-2-7b-chat* [32] and the smaller *Phi-3-mini-4k-Instruct* [2] (3.8 billion parameters). We report the results, along with those of *Llama-3-8b-Instruct* in Figures 13-15.

The performances of both KGW and Aaronson are abysmal on *Llama-2-7b-chat*, with extremely low detectability, even for a high $\delta$ for KGW. At the opposite end, every algorithm performs well on *Phi-3-mini-4k-Instruct*. However, once again, KGW still incurs a high cost in terms of text quality, whereas WaterMax $b = 6$ and Aaronson are both virtually lossless at this regime. *Llama-3-8b-Instruct* sits in between these two extremes.

To understand these results and verify our claim with regard to entropy, we generated $64 \times 100$ completions using each model. Fig. 12 reports the distribution of their entropy. Very clearly, the entropy follows our observation, with *Phi-3-mini-4k-Instruct* having the highest average entropy whereas *Llama-2-7b-chat* has the lowest with a lot more tokens which are close to being deterministic.

This is an essential observation for watermark evaluation since different schemes are better suited to different LLMs. However, WaterMax is currently the only scheme that consistently attains almost perfect detectability at $P_{FA} = 10^{-6}$ irrespective of the LLM, of the temperature, and at almost no cost to text quality.

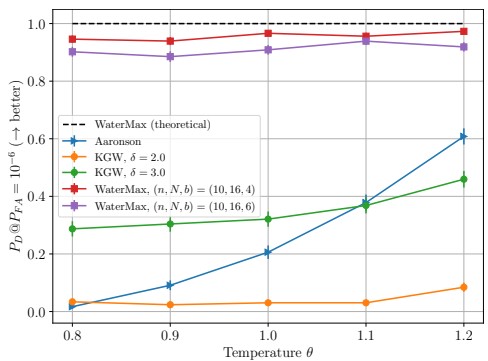
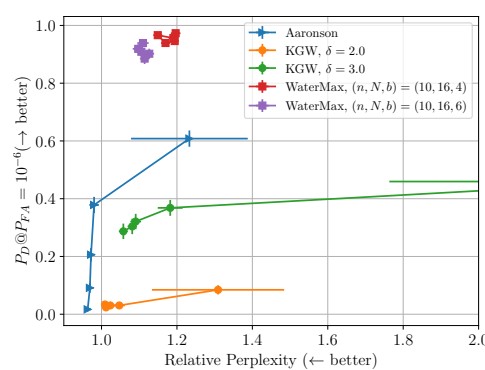

(a) Detectability as a function of the temperature of the LLM.

(b) Detectability as a function of relative perplexity. Each point corresponds to one temperature in the left figure.

Figure 13: Detectability against quality with **Llama-2-7b-chat-hf**, nucleus sampling ($top_p = 0.95$), and hashing window $h = 6$.

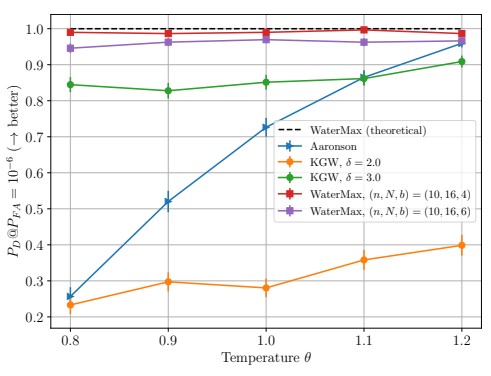
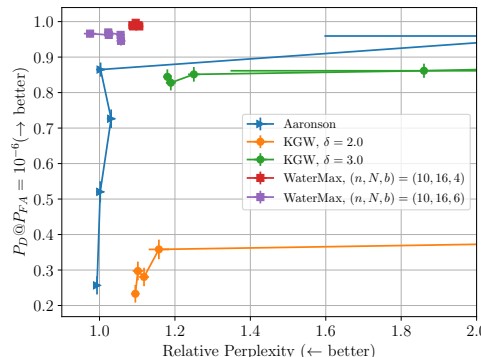

(a) Detectability as a function of the temperature of the LLM.

(b) Detectability as a function of relative perplexity. Each point corresponds to one temperature in the left figure.

Figure 14: Detectability against quality with **Llama-3-8b-Instruct**, nucleus sampling ($top_p = 0.95$), and hashing window $h = 6$.

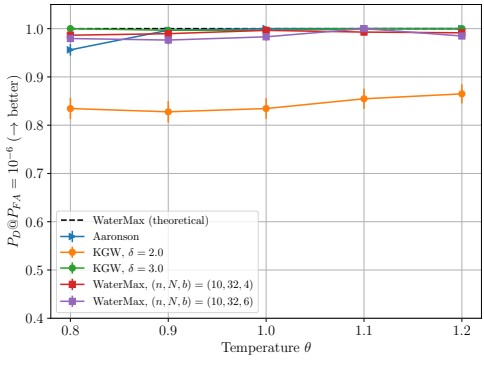
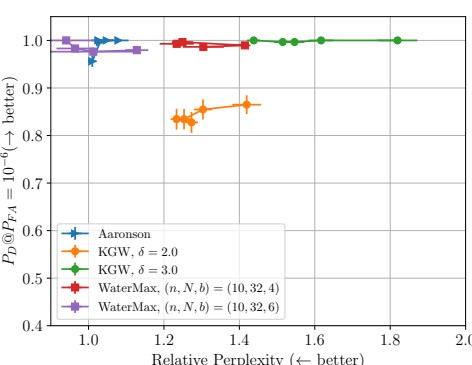

(a) Detectability as a function of the temperature of the LLM.

(b) Detectability as a function of relative perplexity. Each point corresponds to one temperature in the left figure.

Figure 15: Detectability against quality with **Phi-3-mini-4k-instruct**, nucleus sampling ($top_p = 0.95$), and hashing window $h = 6$.

# K  Further experimental results

This section reports more results of interest to the practitioner to tune their watermark of choice. In particular, we report different quality metrics by varying the watermarking parameters. We also study the impact of text size on the detectability of watermarks.

In this appendix, the experiments are based on three MMW tasks defined in [29], namely:

- generating 100 fictional news articles,

- generating 100 reports on well-known books,

- generating 96 stories given a synopsis.

All texts are generated using *Llama-2-chat-hf* with nucleus sampling ($top_p = 0.95$). The hashing window is fixed to $h = 4$ for all algorithms. WaterMax is used with $b = 0$ (no beam-search), which entails a virtually lossless scheme. Every watermarking scheme, except for Aaronson's, is applied on a LLM working at temperature $\theta = 1.0$.

## K.1  Quality

Figures 16a-17c report the relative perplexity using opt-2.7b as an oracle as well as the ROUGE-L score for Aaronson, KGW, and WaterMax as a function of their respective parameter.

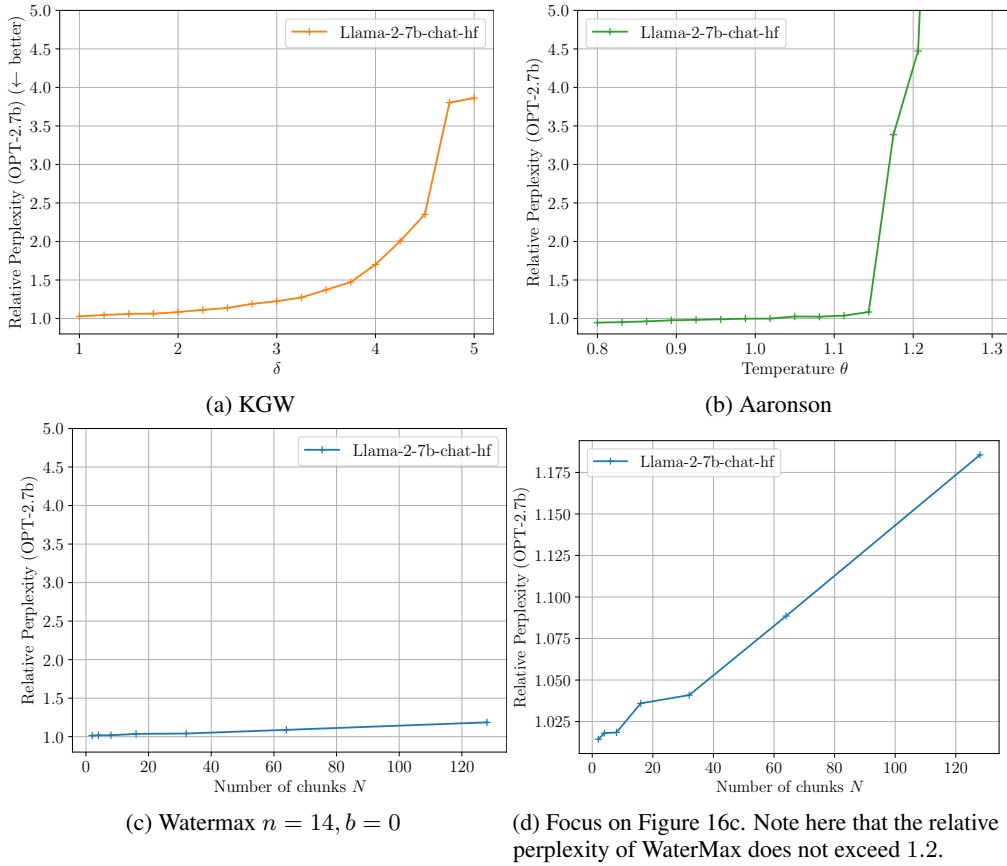

(a) KGW

(b) Aaronson

(c) Watermax $n = 14, b = 0$

(d) Focus on Figure 16c. Note here that the relative perplexity of WaterMax does not exceed 1.2.

Figure 16: Relative perplexity of watermarked text over non-watermarked text measured using opt-2.7b as an oracle.

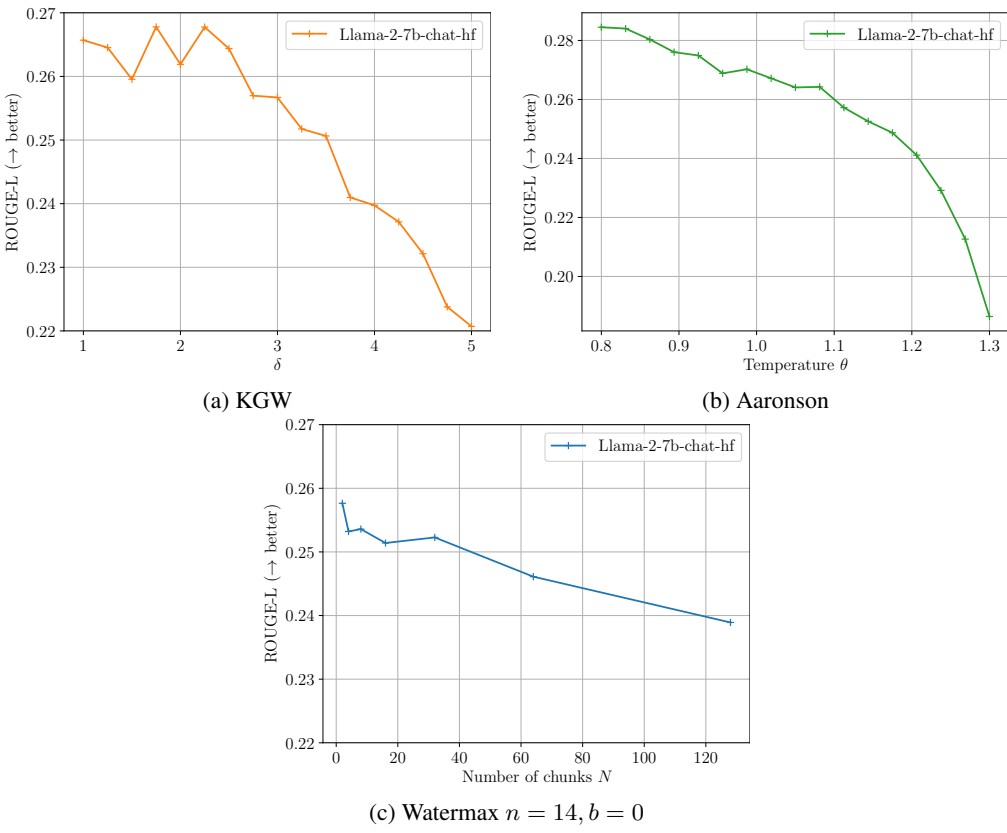

(a) KGW

(b) Aaronson

(c) Watermax $n = 14, b = 0$

Figure 17: ROUGE-L

## K.2 LLM judges

Recent works have proposed using LLM judges to evaluate the quality of watermarked texts. Some examples include the Mark My Words benchmark [29] which directly uses Llama2 as a judge for evaluating watermark quality. Several open-source proxies fine-tuned for this task have been recently proposed in the literature such as Prometheus [20].

Yet, our use of these models for evaluating the quality of watermark texts has not proven fruitful. We provide the results of using Llama-2-7b-chat, Mistral-7b-instruct-v0.2 and Prometheus-7b-v2.0 as LLM judges of the text quality in Fig. 18 of the global response. We asked each of these LLM to grade out of 5 the texts of the "Invented Stories" task of the Mark My Words benchmark following the methodology of [20], using the code provided in the official Github's implementation.

- The grading is inconsistent between LLMs, with Prometheus-7b biased towards higher grades and Llama-2-7b-chat towards lower. Note that the ranking of each watermark is different for each LLM judge. Furthermore, grading can be inconsistent even for non-watermarked texts, with Mistral-7b-instruct-v0.2 highly biased in favor of non-watermarked texts generated with temperature 1.0.

- The average grade does not fluctuate significantly for different watermark strengths, whatever the choice of LLM judge. Interestingly, Fig. 18 shows that increasing $\delta$ can slightly increase the LLM grading, which should not be the case.

- This prompted us to study the grading of barely legible texts, by randomly replacing a percentage of the characters within texts: LLM judges still provide similar average grades despite this attack. More precisely, the grade starts to degrade when the texts starts to look like random strings (around 15% of modified letters). However, there is no impact to the grading for, i.e a percentage of 10%. Here is an example of text with a maximum grade of 5/5: "*sir edward, a chivWlrous knight, had always been driGen by a sense of duty and a*

*thiFst foD SdventuFe. as a yLung man, hW had heard tales of the legendary holy graiP, said to grant the deepest desJreD of FhoWe who posseZsed it. convinced tUat thS grail helX the keT to bringing pFace and prosperity tK hiD kingdLK, sir Wdward set out on a perilLus quesY to find it. he bSgaJ his jouTnFy in the miDty mountaiMs of wales, where he souFht [...]*"

Our conclusion is that, as of writing, these LLM judges don't seem suited to the evaluation of watermarked texts quality. As explained in the paper, we prefer the relative perplexity and ROUGE which fluctuate as expected with watermark strength.

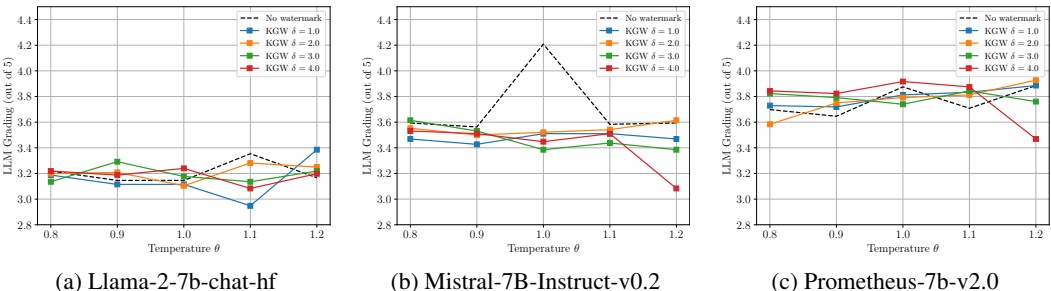

(a) Llama-2-7b-chat-hf   (b) Mistral-7B-Instruct-v0.2   (c) Prometheus-7b-v2.0

Figure 18: Average grade out of 5 for texts generated by Meta-Llama-3-8B-Instruct for the task *"Invented story"* of the Mark My Words benchmark. Texts are watermarked with KGW with different values of $\delta$ and $\gamma = 0.5$. The grade is computed using the Prometheus-eval framework for different LLM models as judges and for different temperatures. The corresponding non-watermarked text is provided to the judge as the reference text.

### K.3 Impact of the text size

We generate texts of a maximum size of $L = 1024$ tokens. For such long texts, the detectability easily reaches $P_D \approx 1.0$ if fixing $P_{FA} = 10^{-6}$. Instead, we fix a detectability of $P_D = 0.5$ and compute the corresponding $p$-values. This metric is close to the definition of 'watermark size' defined by Piet et al. [29]. The advantage of this metric is that we can report an increase in watermark performance even for long texts as $p$-svalues tend to decrease substantially with text size.

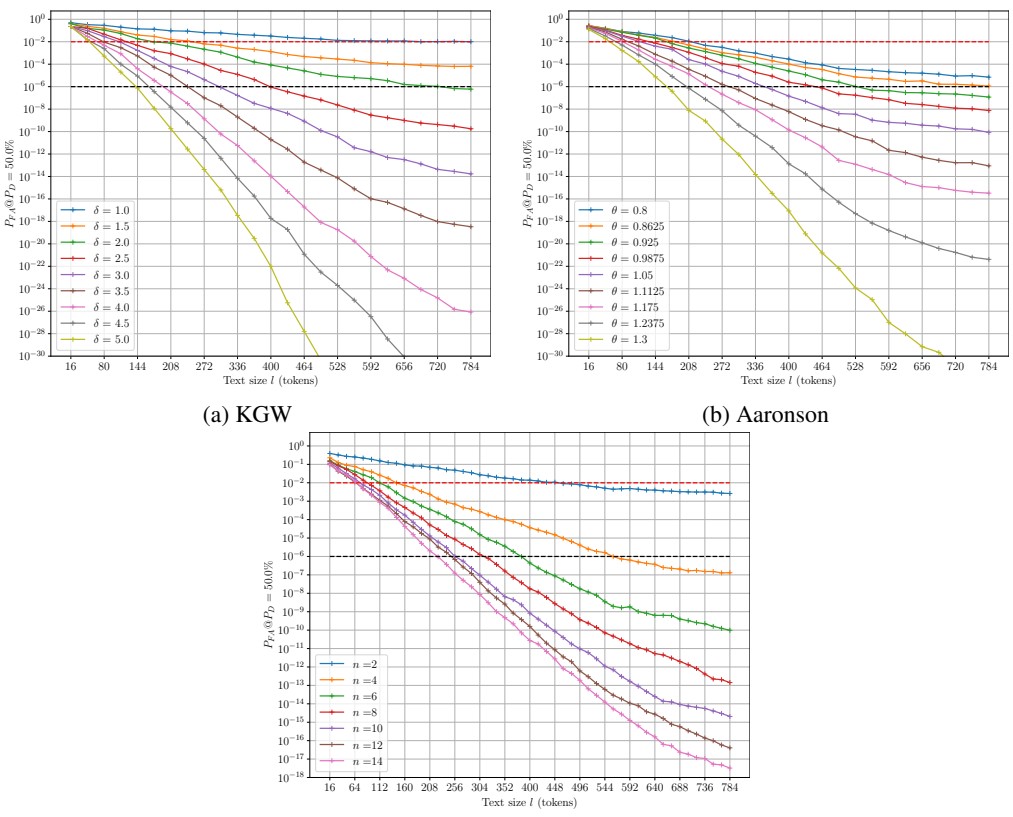

(a) KGW

(b) Aaronson

(c) Watermax ($b = 0$) with chunk size fixed to $l = 16$ tokens. Due to text size not always reaching $1024$ tokens, the effective $N$ is closer to $41$.

