# OpenReview forum: "WaterMax: breaking the LLM watermark detectability-robustness-quality trade-off"
_NeurIPS.cc/2024/Conference — NeurIPS 2024 poster_

### Official Review · Reviewer_5ir5 · 2024-07-04

**Soundness:** 4
**Presentation:** 3
**Contribution:** 4
**Rating:** 7
**Confidence:** 4

**Summary:**

This paper proposes a watermark technique called WaterMax to distinguish LLM-generated texts and human-written texts. WaterMax starts with the watermark detector and asks LLMs to generate a group of candidates from which the one with the lowest p-value determined by that detector is selected as the final output. This work offers a brandnew perspective of text watermarking as there is no formal watermark generator to embed watermark into LLM outputs. Watermax breaks the trade-offs between watermark detectability, quality and robustness, which is carefully discussed in this work with theoretical proof and experiment validation.

**Strengths:**

1. WaterMax offers a brendnew research perspective in the field of text watermarking, where there is no official watermark generator.
2. WaterMax is almost distortion-free as it achieves high text quality on LLM outputs, yet the detectability is preserved.
3. By upgrading the detector, WaterMax can achieve high robustness as well.
4. The superiority of WaterMax is both theoretically proven and experimentally validated.

**Weaknesses:**

1. The dataset used for experiments is limited to high-entropy text generation, yet in reality there is need to watermark LLM-generated codes.

**Questions:**

Is it possible to include low-entropy tasks such as code generation to test if the detector can still properly function? Several works [1,2] can be refered to.

[1].Who Wrote this Code? Watermarking for Code Generation
[2].An Entropy-based Text Watermarking Detection Method

**Limitations:**

It would be better to include more datasets, and further improve the time complexity if possible.

---

> ### Author Rebuttal · Authors · 2024-08-06
>
> > Q1: "Is it possible to include low-entropy tasks such as code generation to test if the detector can still properly function? Several works [1,2] can be refered to.""
>
> We thank the reviewer for the references. At the detection side, these works weight the token value depending on an estimated entropy of the token: hard thresholding for [1], soft weighting for [2]. In principal, this idea can be integrated to WaterMax at the detection side (and thus at the embedding as well since it uses the same score function). Applying hard thresholding [1], WaterMax will not embed a watermark in a chunk where all tokens have low entropy. Yet, we do not recommend the soft weighting of [2]: this prevents from computing a sound p-value.
>
> >[1] *Who Wrote this Code? Watermarking for Code Generation*
> >[2] *An Entropy-based Text Watermarking Detection Method*

---

> > ### Comment · Reviewer_5ir5 · 2024-08-13
> >
> > Thank you for responding. I will maintain my rating.

---

### Official Review · Reviewer_3Sjs · 2024-07-08

**Soundness:** 3
**Presentation:** 3
**Contribution:** 3
**Rating:** 8
**Confidence:** 4

**Summary:**

The authors propose a method for watermarking language models through the use of rejection sampling. By sampling and discarding "chunks" from the model until the p-value returned by an (arbitrary) detection rule is sufficiently low, the proposed method simultaneously preserves output text quality while achieving strong robustness to a variety of attacks. Crucially, the proposed method does not require any intervention within the model itself (e.g. via logit biasing).

**Strengths:**

* The proposed method is both elegant and flexible.
* The paper is clearly written and adequately describes the proposed method.
* The authors demonstrate strong results in terms of text quality, detectability, and robustness against a reasonable selection of attacks.

**Weaknesses:**

* A previous LLM watermarking work, "SemStamp" [1], is similarly based on rejection sampling. Adding experimental comparisons to SemStamp might therefore strengthen the paper by showing how the proposed method fares against the most directly comparable existing method. Otherwise, the authors should probably cite it.
* The authors claim on line 62 that the method of Kuditipudi et al. is the only watermark explicitly designed for robustness; however, the unigram method of Zhao et al. [2] also appears to provide theoretical robustness guarantees.
* While the authors propose an algorithm to limit search over candidate generations, the computational complexity of the proposed method is somewhat concerning. Based on figure 7 (appendix G), it looks like WaterMax incurs ≤ 40% additional runtime on top of generation at a length of only 256 tokens; for the texts in the "Mark My Words" benchmark, the authors report in line 560 that WaterMax requires 5 times the runtime of KGW & Aaronson for watermarked generation. Additional computation is unavoidable with a rejection sampling watermarking scheme, but the authors could address this limitation more clearly within the main paper body.
* A very minor note -- given that the authors consistently compare three watermarking schemes (WaterMax, Aaronson, KGW), it might improve legibility to use a consistent color scheme to refer to these methods across figures.

[1] https://arxiv.org/abs/2310.03991 (NDSS '24)
[2] https://arxiv.org/abs/2306.17439  (ICLR '24)

**Questions:**

* Did the authors explore the use of detection rules other than (11) in conjunction with WaterMax, and if so, did they observe any significant differences in performance?

**Limitations:**

* See comment on efficiency in "Weaknesses" section

---

> ### Author Rebuttal · Authors · 2024-08-06
>
> > W1: "A previous LLM watermarking work, "SemStamp" [1], is similarly ... Adding experimental comparisons to SemStamp ... Otherwise, the authors should probably cite it."
>
> We thank the reviewer for pointing us to SemStamp. We acknowledge that there exists some proximity to our work in the sense that they perform rejection sampling on sentences. It can also be seen as a KGW-type algorithm at the level of sentences instead of tokens.
>
> However, the work exhibits two major weakness that prevent comparison:
> 1) Since the watermark works at the level of sentences and not tokens, extremely large texts are necessary to attain acceptable performance. SemStamp's false positive rate is around 1%. All algorithms studied in our work easily reach 100% TPR within this regime.
> 2) The authors compute the FPR using a z-score which allows an accurate approximation of the p-value only when the number of sentences is high -- however this is never achieved in practice. No empirical validation of this approximation is provided. On the other hand, our work computes exact p-values which are very low for watermarked text. For these reasons, we cannot sensibly compare SemStamp to the state-of-the-art.
>
> > W2: "The authors claim on line 62 that the method of Kuditipudi et al. is the only watermark explicitly designed for robustness; however, the unigram method of Zhao et al. ..."
>
> We agree. We thank the reviewer for pointing out this oversight. We added this reference.
>
>
> > W4: "A very minor note ... it might improve legibility to use a consistent color scheme to refer to these methods across figures."
>
> We corrected the colors of Fig. 1 which were indeed  not consistent with the other figures. Thanks for pointing this out.
>
>
> > Q1: "Did the authors explore the use of detection rules other than (11) in conjunction with WaterMax, and if so, did they observe any significant differences in performance?"
>
> If 'detection rule' means the distribution of the values associated to each token, then yes: As we show in the paper, the choice of detection rule theoretically does not impact the performance of WaterMax.
>
> If detection rules means other kind of score function like SemStamp based semantics, then no, we did not try.

---

> > ### Comment · Reviewer_3Sjs · 2024-08-11
> > **Reply to Authors**
> >
> > I thank the authors for their reply and for the general rebuttal. The additional experiments and promised clarifications re: complexity overhead should strengthen the paper.
> >
> > > We thank the reviewer for pointing us to SemStamp. We acknowledge that there exists some proximity to our work in the sense that they perform rejection sampling on sentences… However, the work exhibits two major weakness that prevent comparison…
> >
> > I’m confused how these weaknesses would prevent at the very least mentioning the prior published sentence-level rejection sampling watermark. If anything they contextualize WaterMax’s strengths.

---

> > > ### Author Response · Authors · 2024-08-12
> > >
> > > I'm sorry, our rebuttal was not clear. We wish to cite this paper due to its similarity, yet we will not include it in the benchmark. We agree that this paper helps contextualize our work. Thanks for your advise.
> > >
> > > Best Regards

---

> > > > ### Comment · Reviewer_3Sjs · 2024-08-12
> > > > **Reply to Authors**
> > > >
> > > > I thank the authors for their clarification. In light of their full rebuttal, which has addressed many of my concerns with the paper, I have raised my review score.

---

### Official Review · Reviewer_F9zP · 2024-07-11

**Soundness:** 3
**Presentation:** 2
**Contribution:** 3
**Rating:** 6
**Confidence:** 4

**Summary:**

The paper presents a novel watermarking scheme for large language models (LLMs). The proposed WaterMax scheme aims to achieve high detectability while maintaining the quality of the generated text, without modifying the LLM's weights, logits, temperature, or sampling technique. WaterMax balances robustness and complexity, distinguishing itself from existing methods that often trade off quality for robustness. The performance of WaterMax is theoretically proven and experimentally validated.

**Strengths:**

1. WaterMax introduces a new detection mechanisms that improve the detectability of watermark in short text, thus preserving the original LLM's token distribution and sampling method.
2. The scheme maintains the quality of the generated text, which is a critical factor in practical applications of LLMs.
3. The paper demonstrates that WaterMax achieves higher robustness and detectability compared to other state-of-the-art watermarking techniques, even under various attack scenarios.
4. The theoretical and experimental evaluations are thorough, covering multiple LLMs and benchmarks. This provides a solid validation of the scheme's effectiveness.

**Weaknesses:**

1. The method involves generating multiple texts for a given prompt and selecting the most suitable one, which increases computational cost and latency. Although the paper suggests ways to limit this, it remains a potential drawback.
2. Based on the randomness in the sampling strategy, LLMs can generate diverse outputs from the same input. This method requires generating multiple outputs and selecting the most suitable one, which may damage the diversity and quality of the generated text.
3. The method's latency, especially in generating longer texts, could be a limitation in time-sensitive applications.
4. The scheme requires careful tuning of parameters to balance robustness, detectability, and computational cost. This adds complexity to its implementation.

**Questions:**

1. How does WaterMax perform with LLMs that exhibit low diversity in generated texts? Are there any measures in place to handle such scenarios effectively?
2. In Line 290, the authors mentioned that there are no methods that can defend against translation attacks. However, many works have been proposed to defend against paraphrase-based and translation-based attacks [1, 2, 3]. Comparison with these methods is necessary.
3. Watermark stealing attack has been proposed recently [4,5,6], which can infer the parameters of the watermarking scheme and remove the watermark from the text. Watermark stealing attacks exhibit significant ability in removing watermark. How does the proposed scheme perform against watermark stealing attacks?
* [1] A. B. Hou et al., “SemStamp: A Semantic Watermark with Paraphrastic Robustness for Text Generation.” http://arxiv.org/abs/2310.03991
* [2] J. Ren et al., “A Robust Semantics-based Watermark for Large Language Model against Paraphrasing.” http://arxiv.org/abs/2311.08721
* [3] Z. He et al., “Can Watermarks Survive Translation? On the Cross-lingual Consistency of Text Watermark for Large Language Models.” http://arxiv.org/abs/2402.14007
* [4] N. Jovanović, R. Staab, and M. Vechev, “Watermark Stealing in Large Language Models.” http://arxiv.org/abs/2402.19361
* [5] Q. Wu and V. Chandrasekaran, “Bypassing LLM Watermarks with Color-Aware Substitutions.” http://arxiv.org/abs/2403.14719
* [6] Z. Zhang et al., “Large Language Model Watermark Stealing With Mixed Integer Programming.” http://arxiv.org/abs/2405.19677

**Limitations:**

This method may affect the quality, diversity and latency of the output text.

---

> ### Author Rebuttal · Authors · 2024-08-06
>
> > W4: "The scheme requires careful tuning of parameters ... This adds complexity to its implementation."
>
> The tuning of our algorithm is **less complex** than the tuning of KGW. The two main parameters of WaterMax are the number of chunk $N$ and the number of drafts per chunk $n$:
> - Detectability/robustness: the performance of Watermax can be computed *a priori* using the theoretical formulas in Eq.(4) or Eq.(6).
> - Quality: $(N,n)$ have basically no impact on text quality.
> - Complexity: Increasing (N,n) increases computational complexity. The cost of $n$ can be reduced through parallelization, wheras the cost of $N$ cannot.
>
> The size of the h-window mostly set the trade-off between robustness and security (against Watermark stealing attack). This is common to any fixed h-window based scheme like KGW and Aaronson's schemes.
>
> In contrast, the choice of $(\gamma,\delta)$ for KGW is not well documented: their impact on the detectability and quality is not easy to predict.  Their setting must be done empirically.
>
>
> >Q1: "How does WaterMax perform with LLMs that exhibit low diversity in generated texts? ... "
>
> A LLM that exhibits low diversity in generated text is a LLM with low entropy in its completions. We direct the reviewer to Appendix J which is explicitly treating this question. We show that both Aaronson's scheme and KGW are sensitive to the LLM's entropy whereas WaterMax 's performance stays constant whatever the choice of LLM and temperature. The reason for this behavior being that WaterMax works on the level of chunks of token whereas the other two algorithms works token per token. This means that, as long as a LLM provides at least some diversity at the chunk level, WaterMax's performance should not suffer compared to other schemes.
>
> > Q2: "In Line 290, the authors mentioned that there are no methods that can defend against translation attacks. However, many works ... [1, 2, 3]. Comparison with these methods is necessary."
>
> We agree with the reviewer: this statement was not correct. We thank the reviewer for providing these references, which we will add to the paper.
>
> However, we cannot compare to the proposed methods as they don't provide any guarantees in terms of false-alarm contrary to the three algorithms studied in our work. Furthermore, due to the lack of theoretical false-alarm rates, these works only compute empirical FPR, --with the papers providing results for FPR from 1% to 10%. This is to be compared to our work which demands false-alarm rates at $10^{-6}$ or below. On the other hand, WaterMax, KGW and Aaronson easily reach 100% TPR with almost not cost on quality or complexity for FPR from 1% to 10%.
>
> > Q3: "Watermark stealing attack has been proposed recently [4,5,6] ... How does the proposed scheme perform against watermark stealing attacks?"
>
> We again thank the reviewer for these references. The problem of watermark stealing is mainly linked to the choice of 1) hashing function and 2) scoring function.
> 1) The attack in [4] only works for the Min-Hash and Sum-Hash functions. We don't use these hashing functions for any watermark scheme as they are intrinsically flawed. The hash is computed recursively on each token in the window, guaranteeing a new hash -- and thus a different key -- for each token. Moreover, we use much longer hash windows: $h=6$.
> 2) The references in [4,5] only work for UNIGRAM and KGW as they are based on the existence of a green-list of tokens. Like Aaronson, WaterMax is not based on a binary partition of the tokens: it associates a *real soft* value to each token. Stealing the secret of this kind of schemes has not been demonstrated.
> 3) References [4,5,6] work token-wise, whereas WaterMax works over chunks, not token. This means that it may select a chunk containing a low valued token because globally that chunk  maximizes the score. Therefore, the frequency of a token may not be related to its associated secret value.

---

> > ### Comment · Reviewer_F9zP · 2024-08-13
> >
> > I thank the authors for answering my questions,  I prefer to maintain the score I initially assigned.

---

### Official Review · Reviewer_VU2G · 2024-07-12

**Soundness:** 3
**Presentation:** 4
**Contribution:** 3
**Rating:** 5
**Confidence:** 4

**Summary:**

The work proposes a new LLM watermarking scheme called WaterMax, that does not modify the distribution or the sampling procedure but uses rejection sampling on multiple generated segments of tokens to cause a high watermark score. A theorem is given to characterize the detector power under attack, given certain independence assumptions. The experiments attempt to demonstrate superior detectability under the constraint of high quality, independent of the temperature/entropy, as well as superior robustness.

**Strengths:**

- **The approach is novel**: the work investigates a fundamentally different approach from prior baselines which is a valuable contribution in itself.
- **Thorough analysis**: the work analyzes the proposed method from several perspectives, and both theoretically and empirically, making it a well-rounded study. I especially appreciate the care taken to discuss the underlying assumptions and investigate their violations in practice.
- **Good writing**: the work is generally very well written, discusses the prior work well, and introduces the method in an understandable way, splitting the different components across different parts of the paper.

**Weaknesses:**

- **Questionable experimental evaluation**: The experimental setup of the main experiments (Figure 5) is mostly reasonable and the observation about the instability of Aaronson holds up. However, the key claim seems to be that "KGW used with common/realistic settings is both of lower quality and has lower detectability than WaterMax". I have several observations here.
1) This claim is made by observing 1.2x relative perplexity measured with a weak 3B model; it is not clear that this is a reliable estimate of text quality in pratical use-cases (see e.g. https://arxiv.org/pdf/2312.02382). The result would be more convincing if a larger model was used, but more importantly, a SOTA LLM (e.g. GPT4) was used as a judge of responses in at least one setting. The author's claim regarding this is ambigious, are they claiming that GPT4 is unreliable for this task?
2) The watermarked model itself is a single 8B model. I see two more small models in Appendix J (7B and 4B). On Llama2-7B the variants of KGW that were tried actually have /higher/ quality that WaterMax, so it is not clear if increasing delta would significantly ruin quality but it may lead to high TPR. Regardless, at least one larger model (e.g., 13B) should in my opinion be tried to substantiate the claim.
3) Even in Fig. 5 a non-standard variant of KGW is used with h=6 and gamma=0.5. To make the claim stronger, can the authors repeat the experiment with more standard h=4 and with gamma=0.25 as well?

- **High computational cost**: while the authors acknowledge this explicitly to some extent, no quantifiable measurement of the computational cost is shown in the main paper, and the appendix suggests that the full experiment took 5x more with WaterMax. This clearly makes WaterMax inapplicable in most real deployments where latency is critical.

However, despite the 5x slowdown, I still believe the paper would have sufficient merit as an exploration of an interesting idea, if the authors could fill the gaps in the experimental evaluation and make the case regarding quality vs power fully convincing. My current borderline accept score is conditioned on the authors providing these additional results to remove my doubts.

**Questions:**

- Is GPT4 unreliable for the task of judging text quality? Can you provide empirical evidence?
- Can you repeat the main experiment with any model larger than 8B?
- Can you repeat the main experiment with a more standard variant of KGW?

**Limitations:**

The only comment I have is regarding quantification of the efficiency degradation which I believe should be explicit in the main paper.

---

> ### Author Rebuttal · Authors · 2024-08-06
>
> > Q1: "*Is GPT4 unreliable for the task of judging text quality? Can you provide empirical evidence?*"
>
>
> First of all, using closed-source models available through API is against our ethics because it prevents reproducibility: GPT-4 is not free; we do not fully control the prompts; it will be no longer available when replaced with GPT-5, etc.
>
> With that said, we acknowledge the trend towards evaluating text quality using LLM judges -- as an example the *Mark My Words* benchmark [1] directly used Llama2 as a judge for evaluating watermark quality. Several open-source proxies fine-tuned for this task have been recently proposed in the literature such as Prometheus [2].
>
> Yet, our use of these models for evaluating the quality of watermark texts has not proven fruitful. We provide the results of using Llama-2-7b-chat, Mistral-7b-instruct-v0.2 and Prometheus-7b-v2.0 as LLM judges of the text quality in Fig. 3 of the global response. We asked each of these LLM to grade out of 5 the texts  of the "Invented Stories" task of the *Mark My Words* benchmark following the methodology of [2], using the code provided in the official Githubs's implementation.
> 1. The grading is inconsistent between LLMs, with Prometheus-7b biased towards higher grades and  Llama-2-7b-chat towards lower. Note that the ranking of each watermark is different for each LLM judge. Furthermore, grading can be inconsistent even for non-watermarked texts, with Mistral-7b-instruct-v0.2 highly biased in favor of non-watermarked texts generated with temperature 1.0.
> 2. The average grade does not fluctuate significantly for different watermark strength, whatever the choice of LLM judge. Interestingly, Fig. 3 shows that increasing $\delta$ can slightly increase the LLM grading, which is a total non-sense.
> 3. This prompted us to study the grading of barely legible texts, by randomly replacing a percentage of the characters within texts: LLM judges still provide similar average grades despite this attack. More precisely, the grade starts to degrade when the texts starts to look like random strings (around 15% of modified letters). However, there is no impact to the grading for, i.e a percentage of 10%. Here is an example of text with a maximum grade of 5/5:
>
> ``
> sir edward, a chivWlrous knight, had always been driGen by a sense of duty and a thiFst foD SdventuFe. as a yLung man, hW had heard tales of the legendary holy graiP, said to grant the deepest desJreD of FhoWe who posseZsed it. convinced tUat thS grail helX the keT to bringing pFace and prosperity tK hiD kingdLK, sir Wdward set out on a perilLus quesY to find it.\n\nhe bSgaJ his jouTnFy in the miDty mountaiMs of wales, where he souFht [...]
> ``
>
> Our conclusion is that, for the moment, these LLM judges don't seem suited to the evaluation of watermarked texts quality.  As explained in the paper, we prefer the relative perplexity and ROUGE which fluctuate as expected with watermark strength were.
>
> [1] *Mark My Words: Analyzing and Evaluating Language Model Watermarks*, J. Piet et al.
>
> [2] *Prometheus: Inducing Fine-grained Evaluation Capability in Language Models*, S. Kim et al.
>
> > Q2: "Can you repeat the main experiment with any model larger than 8B?"
>
> We repeated the experiments with Llama-2-13b and Phi-Medium-4k, two models with 13 and 14 billion parameters respectively. We report the results in the global response in Fig. 1-2. The results are mostly identical to the smaller models: Phi-medium has high entropy and thus all watermarking scheme perform well, whereas Llama-2-13b has low entropy leading to low performance except for WaterMax. This points to the fact that the main parameter of importance is the entropy of the completion, not the size of model. (Note that there is no official 13b version of Llama-3-Instruct.)
>
> > Q3: "Can you repeat the main experiment with a more standard variant of KGW?"
>
> We chose to fix $\gamma = 0.5$ per the recommendations of the *Mark My Words* benchmark [1]. We repeated the experiments following the reviewer's specifications in Fig. 1a and 2a of the pdf. The results are mostly similar and illustrate the trade-off between quality and detectability: smaller $\gamma$ leads to better detectability but lower quality.

---

> > ### Comment · Reviewer_VU2G · 2024-08-12
> >
> > I thank the authors for responding and I acknowledge the provided results as well as the promise to include explicit comments on computation inefficiency to the main paper. My main concern, which is that measuring free text quality using the perplexity of a 3B model is insufficient, was not addressed. I acknowledge the authors' strong ethical stance, and I do not intend to discuss the stance itself further.
> >
> > As the authors acknowledge, using SOTA LLMs as judges of text quality is (for better or worse) a widely popular method in current research, meant to save on costs of human studies. If the authors categorically reject this common evaluation method based on personal beliefs, it is their duty to find another comparably trustworthy way of measuring text quality. One option would be a human study, and another may be testing more capable open models. In their rebuttal response, the authors only experiment with weak 7B models, while there are orders of magnitude more capable open variants. Why were these models ignored?
> >
> > As text quality is a key metric that the work relies on, it is hard to accept the perplexity of a weak 3B model as the only way to measure it.

---

> ### Author Response · Authors · 2024-08-13
>
> We thank the reviewer for their feedback. We would however like to clarify some points:
>
> > If the authors categorically reject this common evaluation method based on personal beliefs, it is their duty to find another comparably trustworthy way of measuring text quality.
>
> Their seems to be a misunderstanding with regard to our stance with respect to evaluating text quality using LLMs. The reason we have not chosen this method to evaluate the performance of watermarking methods is not due to personal belief but, as we claim in the rebuttal, due to its insufficient discriminative power in the specific case of watermarking.
> Evaluating the impact of a watermark on text is a very different problem than say, evaluating the quality of a newspaper article. Indeed, the watermarking signal is (hopefully) weak enough to be hard to detect, and as a consequence, using LLMs as a judge to measure its impact might not be the best tool for the job. Again, the only metrics we have found to vary alongside watermark strength were perplexity and the different variants of ROUGE. Our choice was thus made not due to personal preference but out of necessity of having a meaningful metric in the specific situation of watermarking.
>
> Now, there is an argument to be made against this choice: if an LLM cannot discriminate between watermarked and non-watermarked text, isn't that enough to claim perfect quality preservation?  Such a decision would remove any benefit of so called "distortion-free" algorithms (such as Aaronson's) since even a high $\delta$ of KGW don't seem to impact the grading of an LLM consistently. As an analogy, we could also use LLM judges to grade the overall quality of pictures in the case of image watermarking. Most likely, the problem would be the same as the LLM would likely disregard the slight noise added by watermark methods whereas more common metrics (PSNR, LPIPS, SSIM) would take even small perturbations into account.
>
> The question then becomes: does the watermarker simply wants to preserve content quality as measured by a human judge -- allowing for larger distortion --, or does she want to keep as close to what would be the "natural" distribution of the content? We decided for the latter as it seems to capture the impact of watermark on text better than grading.
>
> >In their rebuttal response, the authors only experiment with weak 7B models, while there are orders of magnitude more capable open variants. Why were these models ignored?
>
> We followed the methodologies proposed recently in the  watermarking literature: the *Mark My Words* benchmark [1] claims good grade correlations between GPT-3.5 and the Llama-2-7b model (see Section 4.1 of their paper where they claim a $R^2 =0.97$). Furthermore, the 7b model of Prometheus-eval [2] is claimed to attain SOTA performance at its size and is specifically fine-tuned on GPT-4 evaluations. If we are to trust the results from these works, there should not be a large gap in grading between model sizes.
>
> > [It] is hard to accept the perplexity of a weak 3B model as the only way to measure [text quality].
>
> We chose Opt-2.7b for measuring perplexity as to follow KGW methodology in [3] (see Section 6 of their paper). We haven't found any difference between using a small or a larger model  for measuring perplexity. In our case we tested Llama-2-7b, Llama-3-8b and Mistral-7b-v0.2 and despite some differences in the absolute value of the perplexity measured, there was no difference in relative perplexity between the watermarking algorithms -- which is what we want to measure. Consequently, we opted for the smaller model in order to match with the methodology of previous art.
>
>
> [1] _Mark My Words: Analyzing and Evaluating Language Model Watermarks_, J. Piet et al.
>
> [2] _Prometheus: Inducing Fine-grained Evaluation Capability in Language Models_, S. Kim et al.
>
> [3] _A watermark for large language models_ Kirchenbauer, John, et al.

---

### Author Rebuttal · Authors · 2024-08-06

We thank all the reviewers for their constructive comments which help us improving our submission.

**The reviewers globally find that the limitations on the complexity is not enough outlined.**

The submission already accounted that:
- The main point of WaterMax is to strike the trade-off detectability/complexity (without loss of quality) instead of the trade-off detectability/quality (without extra complexity) well documented in the literature.
- The paper proposes means to reduce the complexity overhead and latency (from sampling text to chunks, sampling chunks in parallel, beam search ...).
- App. G shows experimental results about runtimes.

$\to$ *We agree that the complexity overhead and the latency of the final scheme is not properly reported in the main body. We will rewrite Sect. 7.4 "Computational complexity" explicitly as a limitation with reported runtime and latency measurements. Especially, we will clearly say that WatermMax is 5 times slower for our recommanded setup.*

**The attached pdf file shows the following news results as suggested by the reviewers.**

- Fig. 1.a and Fig. 2.a: repeat the main experiment with a standard KGW set as $(\gamma,h)=(0.25,4)$ as Reviewer VU2G suggested.
- Fig. 1.bc and Fig 2.bc: repeat the main experiment with larger models (Llama-2-13b and Phi3-medium), as Reviewer VU2G suggested.
- Fig 3: demonstrate the limited applicability of LLM judges in evaluating watermarked text quality, in response to Reviewer VU2G's question.

---

### Decision · Program_Chairs · 2024-09-25

**Decision:**

Accept (poster)

**Comment:**

This paper proposes a new LLM watermarking scheme, WaterMax, which achieves high detectability while maintaining the quality of generated text. The performance of WaterMax is theoretically proven and experimentally verified. LLM watermarking is a less-regarded but interesting research topic. The proposed watermarking scheme is novel, and the theoretical analysis and experimental evaluation are thorough. The paper is generally well-written. In the rebuttal stage, the authors addressed most of the issues raised by the reviewers. The next version needs to properly report the complexity overhead and latency of the scheme.